# Dynamics of tissue repair regulatory T cells and damage in acute *Trypanosoma cruzi* infection

Santiago Boccardo[1,2], Constanza Rodriguez[1,2], Camila M. S. Gimenez[1,2], Cintia L. Araujo Furlan[1,2], Carolina P. Abrate[1,2], Laura Almada[1,2], Manuel A. Saldivia Concepción[3], Peter Skewes-Cox[3], Srinivasa P. S. Rao[3], Jorge H. Mukdsi[4,5], Carolina L. Montes[1,2], Adriana Gruppi[1,2], Eva V. Acosta Rodríguez [1,2]*

1 Centro de Investigaciones en Bioquímica Clínica e Inmunología (CIBICI-CONICET). Córdoba, Argentina, 2 Departamento de Bioquímica Clínica, Facultad de Ciencias Químicas, Universidad Nacional de Córdoba. Córdoba, Argentina, 3 BioMedical Research, Novartis, Emeryville, California, United States of America, 4 Instituto de Investigaciones en Ciencias de la Salud (INICSA-CONICET). Córdoba, Argentina, 5 Centro de Microscopia Electrónica, Facultad de Ciencias Médicas, Universidad Nacional de Córdoba. Córdoba, Argentina

* eva.acosta@unc.edu.ar

**Data Availability Statement:** The datasets generated for this study can be found in the NIH repository under accession number PRJNA941341 (https://www.ncbi.nlm.nih.gov/sra/PRJNA941341).

## Abstract

Tissue-repair regulatory T cells (trTregs) comprise a specialized cell subset essential for tissue homeostasis and repair. While well-studied in sterile injury models, their role in infection-induced tissue damage and antimicrobial immunity is less understood. We investigated trTreg dynamics during acute *Trypanosoma cruzi* infection, marked by extensive tissue damage and strong CD8+ immunity. Unlike sterile injury models, trTregs significantly declined in secondary lymphoid organs and non-lymphoid target tissues during infection, correlating with systemic and local tissue damage, and downregulation of function-associated genes in skeletal muscle. This decline was linked to decreased systemic IL-33 levels, a key trTreg growth factor, and promoted by the Th1 cytokine IFN-γ. Early recombinant IL-33 treatment increased trTregs, type 2 innate lymphoid cells, and parasite-specific CD8+ cells at specific time points after infection, leading to reduced tissue damage, lower parasite burden, and improved disease outcome. Our findings not only provide novel insights into trTregs during infection but also highlight the potential of optimizing immune balance by modulating trTreg responses to promote tissue repair while maintaining effective pathogen control during infection-induced injury.

## Author summary

During Chagas' disease, caused by the protozoan *Trypanosoma cruzi*, severe organ damage is generated by the interplay between the parasite and the immune response. In our investigation, we examined the role of tissue-repair regulatory T cells (trTregs) during the acute phase of *T. cruzi* infection in mice. Surprisingly, we observed a reduction in trTregs at the peak of tissue damage, contrary to their usual accumulation after injury in other

**Funding:** Research reported in this publication was supported by the Agencia Nacional de Promoción de la Investigación, el Desarrollo Tecnológico y la Innovación (https://www.argentina.gob.ar/ciencia/agencia) (PICT 2020-SERIE A - 0487 to EVAR) and the National Institute of Allergy and Infectious Diseases of the National Institutes of Health (https://grants.nih.gov/grants/funding/r01.htm) (R01AI169482 to EVAR). The funders had no role in study design, data collection and analysis, decision to publish, or preparation of the manuscript.

**Competing interests:** Manuel A. Saldivia Concepción, Peter Skewes-Cox and Srinivasa P. S. Rao are employees of Novartis and hold shares of Novartis. The rest of the authors have declared no competing interest.

contexts. This decline aligned with decreased levels of interleukin-33, a critical factor for trTreg survival, and was promoted by the effector cytokine IFN-γ. Administering interleukin-33 at early infection times not only boosted trTregs but also expanded other reparative and antiparasitic immune cells. Consequently, these treated mice exhibited reduced damage and lower parasite levels in tissues. Our findings provide new insights into how trTreg function during infection-related injury, paving the way for strategies that balance the immune response to support tissue repair without weakening the body's ability to fight the infection. This approach could have broader implications for treating infectious diseases and conditions involving tissue damage.

## Introduction

Regulatory T cells (Tregs) are CD4+ T lymphocytes with immunoregulatory properties characterized by the expression of the transcription factor Forkhead box P3 (Foxp3) [1]. The immunomodulatory role of Tregs has been extensively described across diverse biological processes, encompassing tolerance maintenance, autoimmunity and allergy, cancer, infections and immunometabolic diseases [2]. This wide spectrum of activities underscores the adaptability of Tregs in regulating various effector responses, including Th1, Th2, and Th17 immunity. This phenomenon, recognized as Tregs specialization, enables them to tailor their regulatory capacity to distinct scenarios and specific immune profiles [3].

In recent decades, it has become evident that Tregs residing in non-lymphoid tissues are a heterogeneous population and perform roles that go beyond their classical suppressive functions, both during homeostasis and in response to injury [4–6]. Tissue Tregs adapt to their locations to perform tissue-specific functions, such as regulating metabolism and restraining obesity in adipose tissue [7], orchestrating tissue regeneration and homeostasis in the colon [8], aiding in skeletal muscle (SM) regeneration and controlling fibrosis [9,10], facilitating wound healing and hair growth in the skin [11,12], and promoting myelin regeneration in the central nervous system (CNS) [13]. Although originally thought to seed tissues at final differentiation, recent work suggests that all Tregs, including tissue Tregs, can recirculate among non-lymphoid tissues and undergo transient residency and adaptation in response to antigenic signals [14].

Within tissue Tregs, a specialized subset known as tissue repair Tregs (trTregs) has been identified [15]. TrTregs exhibit a highly activated state, sharing a phenotypic and transcriptomic core signature across different tissues, while also possessing unique features shaped by their microenvironment. These cells are equipped to play key roles in maintaining tissue homeostasis and promoting repair following damage [15,16]. Although initially believed to reside only in non-immune tissues, trTregs have also been found in lymphoid organs. The development of the trTreg program occurs through a stepwise process, starting with an initial commitment in peripheral lymphoid organs [17–19]. At each stage, trTreg survival, expansion, and acquisition of tissue-repair properties depend on IL-33, an alarmin from the IL-1 family released during cellular damage [20]. Correspondingly, trTregs express the specific IL-33 receptor subunit, ST2 [15,18].

Most studies on ST2+ Tregs have focused on their behavior under homeostatic conditions or in models of sterile injury, where they accumulate locally to facilitate tissue healing [21]. However, there is limited information on the behavior of this subset and their reliance on the IL-33/ST2 axis for controlling infection-induced tissue damage. The few studies available have primarily evaluated total tissue Tregs with some specifically analyzing the trTreg subset. These studies demonstrate a local accumulation of Tregs and highlight diverse roles in promoting

tissue repair during acute infections, such as those caused by influenza virus [22], herpes simplex virus [23], and cytomegalovirus [24]. Interestingly, Tregs accumulation in the lung and cornea appears to depend more on IL-18 than IL-33 signaling, suggesting the involvement of a different subset of reparative Tregs. In helminth infections, Tregs increased in the liver during *S. japonicum* infection, significantly ameliorating tissue pathology [25]. Conversely, in chronically HIV-infected patients, ST2+ Tregs in the intestinal lamina propria appear to have a limited role in tissue repair, as evidenced by increased epithelial permeability and tissue fibrosis [26]. In protozoan infections, tissue Tregs play protective roles in cerebral malaria [27], but their relevance in Toxoplasma-associated SM pathology seems negligible [28].

Despite these findings, the specific role of trTregs and the IL-33/ST2 axis in tissue repair and homeostasis during infections remains poorly understood. Furthermore, few studies have directly examined how specifically manipulating trTreg abundance affects antimicrobial immunity and pathogen load control in tissues. Given the suppressive potential of tissue- and IL-33-expanded Tregs in certain conditions [4,29], further research is needed to better understand their influence on effector immune responses. This will enhance our understanding of the interplay between microbial persistence, tissue damage and repair, as well as the immunopathology of chronic infections.

Chagas' Disease (American Trypanosomiasis) is a chronic infection caused by the protozoan parasite *Trypanosoma cruzi*. Endemic to Latin America, it also affects non-endemic regions, with approximately 6–7 million people infected worldwide [30]. During the acute phase, the parasite invades various tissues, including muscle, liver, gut, lymph nodes, spleen, and CNS, where it replicates and causes cell death and tissue damage. This phase is associated with high parasitemia and nonspecific symptoms. Type 1 immunity, characterized by elevated proinflammatory cytokines like IFN-γ [31–33], and innate and adaptive immune cells such as NK cells, inflammatory macrophages, and CD8+ T lymphocytes [32,34,35], work to reduce parasite replication and burden. However, the immune response cannot completely eradicate the parasite, leading to chronic infection. In the chronic phase, about 30% of infected individuals develop digestive or cardiac symptoms after 10–30 years [36]. Evidence suggests that both parasite persistence and sustained inflammation mediate the tissue damage characteristic of the chronic phase [37]. Additionally, muscular pain and weakness are common in both acute and chronic Chagas' patients [38–40]. An association between SM parasitism and myositis, with structural alterations of muscle fibers, has been demonstrated in chronically infected humans [41,42] and in mouse models of acute and chronic infection [43–47].

In this study, we used a mouse model of acute *T. cruzi* infection to investigate trTregs and their role in tissue damage and protective immunity. We examined trTreg dynamics in this infection scenario, which is characterized by a unique combination of elevated systemic tissue injury and a limited Tregs response [48]. We found reduced trTreg numbers in target tissues that correlated with lower systemic IL-33 levels. By supplementing with recombinant IL-33, we evaluated the impact of trTregs and other IL-33-responsive immune subsets on tissue damage, parasite control, and infection progression. Our work highlights the balance between microbicidal and regenerative responses driven by trTregs and the IL-33/ST2 axis, which is still not well understood in infections and may be particularly relevant to the progression of acute Chagas' Disease.

## Results

### Systemic tissue damage during acute *T. cruzi* infection associates with tissue parasitism and immune infiltration

To explore tissue injury and repair in our acute *T. cruzi* infection model, we first assessed the extent and progression of damage. Intraperitoneal injection of 5,000 trypomastigotes

(Tulahuen strain) in Foxp3 reporter mice induces a mortality rate of approximately 50%, occurring within a narrow window during the acute phase of infection, typically between 20–22 and 35–38 days post infection (dpi) (Fig 1A). This mortality window is preceded by a significant weight loss (S1A Fig), a peak of parasitemia (Fig 1B) and high parasite load in tissues such as SM, heart, spleen and liver (Fig 1C) around 21 dpi. These events are accompanied by the peak of immune cell expansion in the spleen (Fig 1D). In agreement with previous findings [43,49], histological examination of SM at 21 dpi revealed the presence of parasite nests and diffuse mononuclear infiltrate associated with necrosis and dystrophic calcification of muscular fibers, absent in samples from non-infected (NI) mice (Fig 1E). Quantification of the immune infiltrate in SM showed that leucocyte counts were also at their maximum at 21 dpi (Fig 1F).

To further elucidate the impact of *T. cruzi* infection on muscle physiology, we conducted a whole-tissue RNAseq comparing infected (INF) versus NI quadriceps. The transcriptome analysis identified 1621 differentially expressed genes (DEGs) between both conditions. Non-supervised pathway analysis of the DEGs using EnrichR revealed that, in addition to pathways associated with immune responses such as interferon-gamma, interferon-alpha and complement responses, several pathways related to muscle physiology such as oxidative phosphorylation, myogenesis and adipogenesis were among the most significantly enriched pathways (Fig 1G). As expected, volcano plots showed that most genes associated with the pathways linked to immune responses including "Interferon gamma, Interferon alpha and Complement" were upregulated by the infection (S1B Fig and S1 Table). In contrast, the majority of genes linked to muscle physiology pathways such as "Oxidative Phosphorylation, Myogenesis and Adipogenesis" were downregulated (Fig 1H and S1 Table), supporting the notion that acute infection disrupted SM homeostasis. Consistent with histological and transcriptomic evidences of SM damage, we found increased plasma activity of creatine phosphokinase (CPK) and creatine phosphokinase of muscle and brain (CPK-MB) at 21 dpi compared to NI mice (S1C Fig). The alteration of additional markers of systemic damage such as increased activity of lactate dehydrogenase (LDH), glutamic oxaloacetic transaminase (GOT) and glutamic pyruvic transaminase (GPT), along with hypoglycemia, indicated affection of various target tissues beyond SM, as previously reported by our group [48,50]. As expected, the greatest tissue alterations coincide with highest parasitemia levels and tissue parasitism (Fig 1B and 1C), as well as with the peak of immune cells expansion in the spleen (Fig 1D) and maximum immune infiltration in SM (Fig 1F) and other target tissues, such as heart and liver (S1D Fig). Given these features, the acute phase of *T. cruzi* infection emerged as an instrumental setting to study trTreg roles during the infection.

## *Bona fide* trTregs are reduced in target tissues and lymphoid organs at the peak of infection

To understand the dynamics of reparative regulatory T cell responses during the acute phase of *T. cruzi* infection, we first evaluated the presence of total Tregs and trTregs in various parasite target sites using flow cytometry. As previously documented [48], *T. cruzi* infected mice exhibited reduced frequencies of total Tregs in spleen and liver around the peak of the infection (21 dpi) compared to NI controls (S2A and S2B Fig). An even more pronounced decrease was observed when analyzing other peripheral tissues known to be common parasite targets, such as SM and the heart. We then quantified trTregs from SM, liver and spleen, identified as proposed by Delacher et al. [15], by their co-expression of ST2 and KLRG-1 in the Foxp3+ Tregs population (Fig 2A). Kinetic studies on trTregs showed a decrease in their frequency in SM, liver, and spleen during infection, with statistically significant reduction observed at

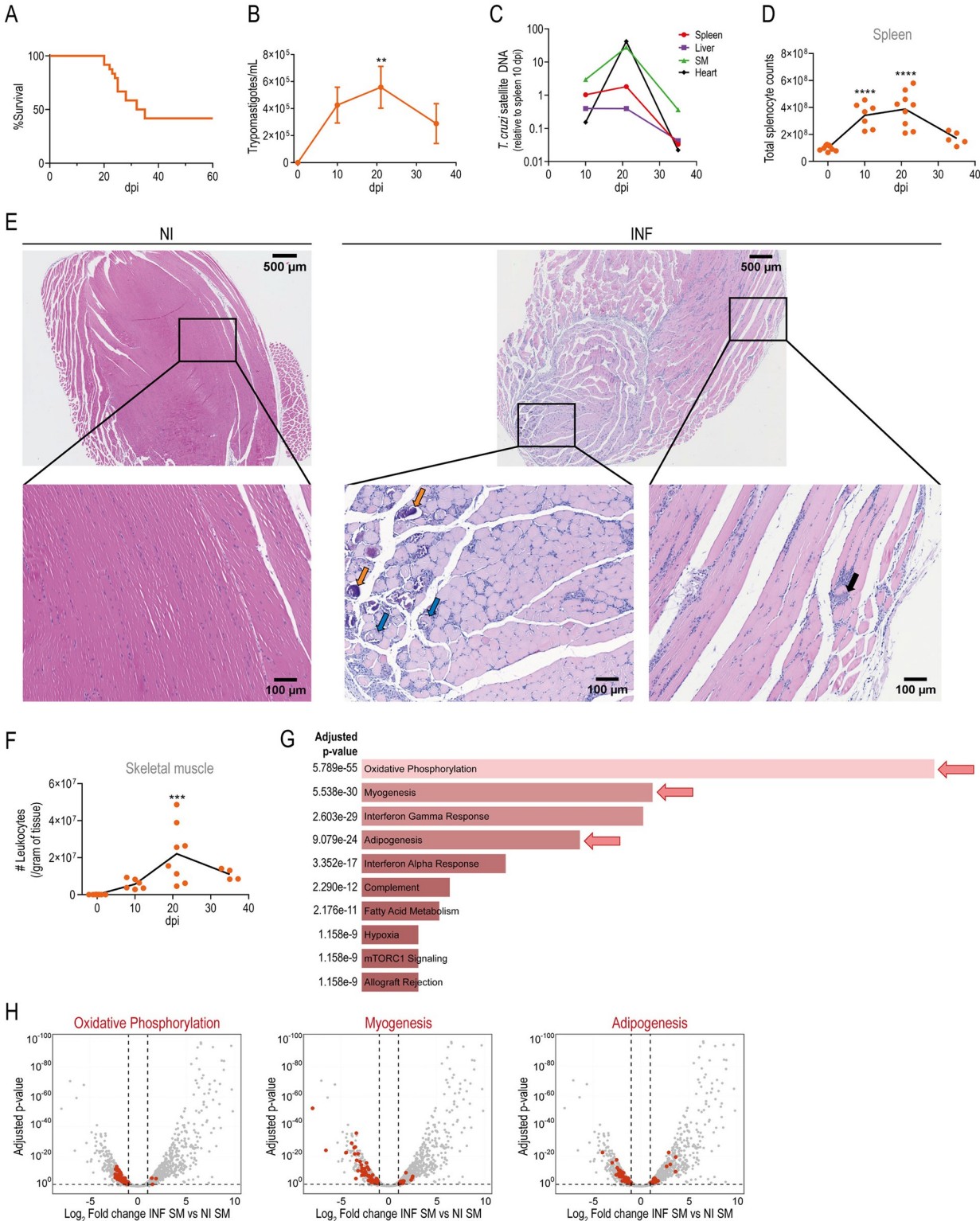

**Fig 1. Characterization of muscle and systemic tissue damage during acute *T. cruzi* infection.** Foxp3-GFP mice were infected with *T. cruzi* and infection progression and tissue damage were evaluated at different days post infection (dpi). (A) Survival curve; N = 24. (B) Kinetics of parasite counts in blood. (C) Kinetics of *T. cruzi* satellite DNA quantification in skeletal muscle (SM), heart, liver and spleen. (D) Kinetics of total splenocyte counts. (E) Representative Hematoxylin-Eosin stain of quadriceps muscle from non-infected (NI) and infected (INF) (21 dpi) mice; N = 4–7 per group. Black arrow: parasites nest, blue arrows: necrotic muscle fibers, orange arrows: muscle fiber with dystrophic calcification.

Magnification: Upper row = 1.9X, bottom row = 10X. (F) Kinetics of SM leukocyte counts normalized to tissue weight. (G and H) Whole quadriceps SM RNAseq data analysis from NI and INF animals; N = 3 per group. (G) Non-supervised pathway analysis of the differentially expressed genes between INF and NI SM. Bars show the top-ten pathways upregulated in INF SM with red arrows highlighting pathways related to SM physiology. (H) Volcano plots displaying differentially expressed genes between INF and NI SM. According to (G), genes associated with oxidative phosphorylation (left), myogenesis (center) and adipogenesis (right) pathways are highlighted in red. (B) Data are presented as mean ± SEM; N = 7–23 per dpi. (C) Data are presented as mean and values are normalized to spleen parasitism at 10 dpi; N = 3 per dpi. (D and F) Data are presented as individual values (circles) and mean (line). (B, D and F) Statistical significance was determined by one-way ANOVA. P values are relative to 0 dpi: **p < 0.01; ***p < 0.001; ****p < 0.0001. (A-F) Data were collected from 2–3 independent experiments.

21 dpi (Fig 2B). Absolute numbers of trTregs were also significantly reduced at this time point across all evaluated tissues (Fig 2B). Although their counts tended to recover at later time points, we did not observe the accumulation seen in other injury models [21–24]. The trTreg subset was not assessed in the heart due to the low infiltration of total Tregs.

We then evaluated whether trTregs, identified by co-expression of ST2 and KLRG-1, exhibited the distinctive phenotype of tissue repair cells during infection, distinguishing them from classic ST2- KLRG-1- lymphoid-like Tregs as described in other settings [15,18,51–55]. Using flow cytometry, we compared the expression of transcription factors (BATF, IRF4, and Ki-67), surface molecules (CD44, CD62L, TIGIT, ICOS, GITR, PD-1, and CTLA-4), and amphiregulin (Areg) in trTregs and ST2- KLRG-1- Tregs from the spleens of INF and NI mice, identified as shown in Fig 2A. According to the representative dot plots from S3 Fig and summarized in the heatmap in Fig 2C, spleen trTregs from NI mice exhibited higher expression levels of most markers, except CD62L, compared to ST2- KLRG-1- Tregs from the same mice. Notably, trTregs from INF mice also displayed a *bona fide* trTreg phenotype, as defined by Delacher et al [15], with higher expression of markers like BATF, IRF4, CD44, TIGIT, ICOS, GITR, PD-1, CTLA-4 and Areg compared to their ST2- KLRG-1- counterparts. Principal component analysis (PCA) of the phenotypic data showed that while ST2- KLRG-1- Tregs from NI and INF mice clustered closely, trTregs segregated apart along PC1, which explains around 70% of the variance (Fig 2D). Additionally, trTregs from NI and INF mice differed along PC2, accounting for 20% of the variance, mainly driven by the stronger expression of different markers, particularly CD62L, in trTregs from INF mice.

In summary, our results indicate that during acute *T. cruzi* infection, despite the severe systemic tissue damage, trTregs are particularly reduced within an already restricted total Tregs pool. Notably, the few remaining trTregs found at the peak of infection retain key features of *bona fide* trTregs while exhibiting distinctive characteristics, including higher expression of some markers, likely shaped by the infection context.

## Systemic IL-33 levels are reduced during acute *T. cruzi* infection

It is established that trTreg development is a multi-step process that initiates with a specialization commitment in secondary lymphoid organs and culminates after migration into residence tissues [17–19]. Remarkably, IL-33 has been shown to play a role in all these different stages [5]. Given the decline in trTreg numbers despite elevated tissue damage at the peak of acute infection, we assessed IL-33 concentrations at systemic and peripheral sites. Our analysis showed that plasma IL-33 concentration was detectable in NI mice but diminished during the course of acute *T. cruzi* infection, with a statistically significant decrease at 21 dpi, followed by a return to baseline levels at 35 dpi (Fig 3A). A similar trend was observed in spleen homogenates (S4A Fig). In contrast, IL-33 levels in SM increased at 21 and 35 dpi (Fig 3A), while remained constant in the liver (S4A Fig). These results indicate that despite a conservation or increase in IL-33 levels in peripheral non-lymphoid tissues, its concentration is reduced at the

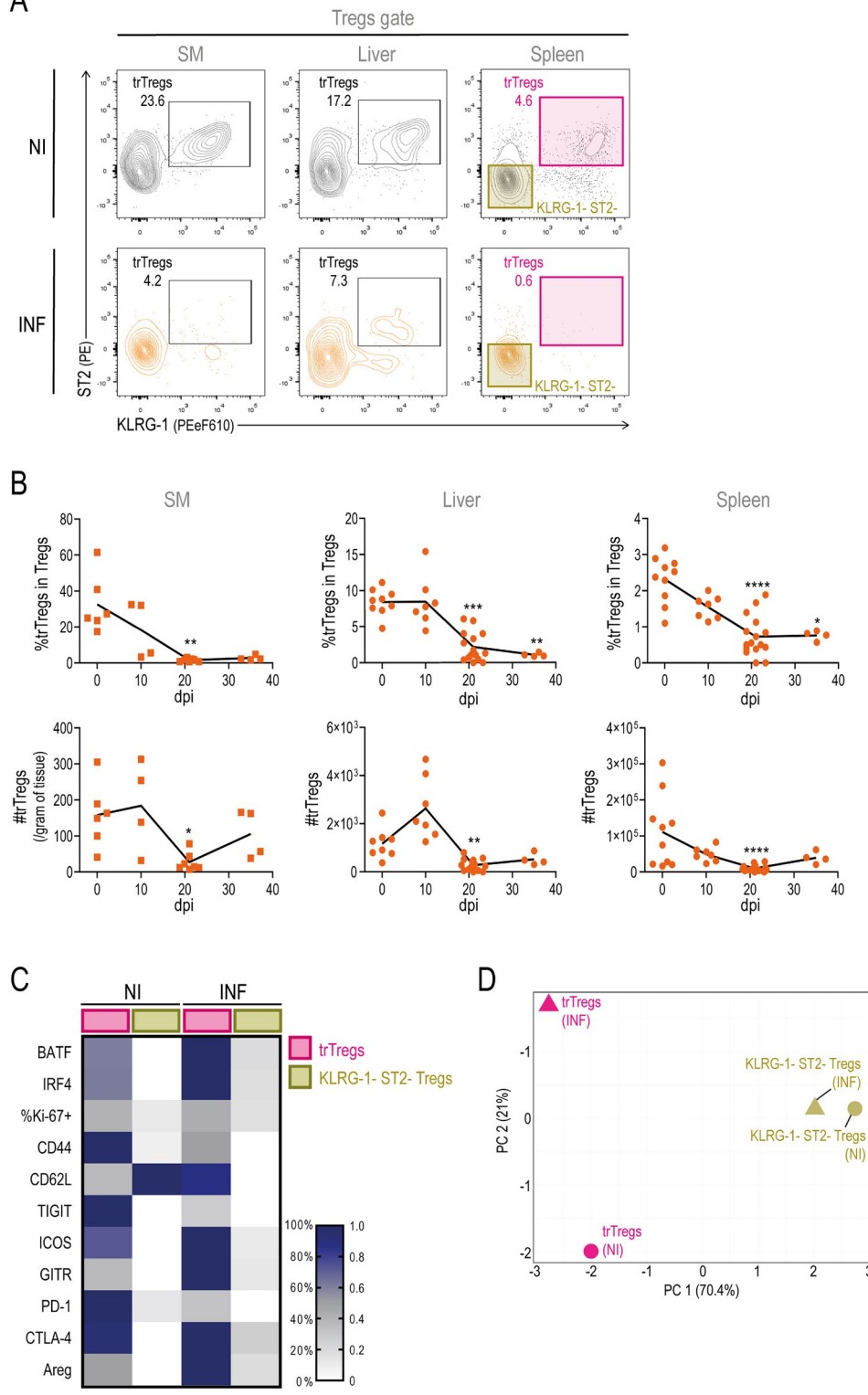

**Fig 2. Tissue repair Tregs are reduced in target tissues and lymphoid organs during acute *T. cruzi* infection.**
Tissue repair Tregs (trTregs) were studied by flow cytometry in tissues from *T. cruzi* infected Foxp3-GFP mice at different days post infection (dpi). (A) Representative dot plots showing ST2 and KLRG-1 staining in total Tregs present in skeletal muscle (SM), liver and spleen obtained from non-infected (NI) and infected (INF) (21 dpi) mice. ST2+ KLRG-1+ cells (pink gate) were defined as trTregs. (B) Kinetics of trTreg frequency within total Tregs (upper

row) and absolute number (bottom row) in SM, liver and spleen. For SM, squares represent values obtained from pools of 2–4 mice and cell counts are normalized to tissue weight. For liver and spleen, circles represent values from individual mice. For all tissues, the mean of individual values is shown by a line. Statistical significance was determined by Kruskal-Wallis test. P values are relative to 0 dpi: *$p < 0.05$; **$p < 0.01$; ***$p < 0.001$; ****$p < 0.0001$. (C) Heatmap displaying the relative expression or frequency of the indicated markers in splenic trTregs (pink gate) or ST2- KLRG-1- Tregs (golden gate) as defined in (A), evaluated in NI and INF animals, as presented in S3 Fig. (D) Principal component analysis of data presented in (C) without relativizing. (A-D) Data were collected from 2–3 independent experiments.

systemic level and in secondary lymphoid tissues, suggesting that the initial steps of trTreg development may be affected during *T. cruzi* infection.

## IL-33 supplementation expands trTregs from acutely infected animals *in vitro* but fails to prevent trTreg reduction in established infection

As discussed in the previous section, reduced systemic IL-33 levels during infection may compromise trTreg commitment in secondary lymphoid organs, leading to an overall decrease in trTreg numbers. Therefore, we hypothesized that supplementing with recombinant IL-33 could increase trTreg counts in our experimental setting and provide clues about the role of trTregs in managing tissue damage and pathology progression during *T. cruzi* infection. To test this hypothesis, we first evaluated the responsiveness of trTregs from INF mice to IL-33 *in vitro*. Splenic Tregs (CD4+ Foxp3-GFP+) sorted from NI and INF mice were stimulated with anti-CD3, anti-CD28 and recombinant IL-2 in the presence or absence of IL-33. The addition of IL-2 and IL-33 resulted in increased frequencies of ST2+ KLRG-1+ trTregs, not only in cultures of Tregs from NI mice, as previously reported [8,29], but also in Treg cells obtained from INF mice (Fig 3B). The IL-33-mediated expansion of trTregs was confirmed by cell counts, which showed an increase only in this subset and not in total Tregs (S4B Fig). Notably, IL-33 supplementation failed to induce ST2 and KLRG-1 expression in conventional T cells (CD4 + Foxp3-GFP-) obtained from the spleens of either NI or INF animals (S4C Fig). These *in vitro* results provide a proof of principle that IL-33 targets the Tregs pool by expanding trTregs, even when present at low frequency, as observed in *T. cruzi* infection environments, supporting its potential for treatment of INF animals.

For the *in vivo* treatment, aimed at preventing the infection-induced reduction of trTregs, we administered IL-33 (or PBS as control) intraperitoneally (i.p.) at 12, 15, and 18 dpi, as shown in Fig 3C. Alternatively, mice were co-administered with IL-2 and IL-33, considering the relevance of IL-2 for Tregs survival [56] and noting that systemic levels of this cytokine remain unchanged despite increased cell demand during acute *T. cruzi* infection [48]. As a positive control, we observed that the *in vivo* administration of IL-33 to NI animals led to trTreg expansion (Fig 3D), as previously reported [15]. Remarkably, treatment with IL-33 or IL-33 plus IL-2 in INF mice failed to increase trTreg frequency or absolute numbers in SM, spleen, or liver compared to PBS-treated controls, and these levels remained significantly lower than those observed in NI mice (Fig 3E). To further assess whether IL-33 supplementation impacted tissue damage or infection progression, despite its limited effect on trTregs, we examined plasma levels of biochemical markers indicative of tissue damage. IL-33 treatment did not improve these markers, and IL-2 plus IL-33 appeared to worsen them compared to PBS-treated controls (Fig 3F). Additionally, global indicators of disease progression, such as total weight loss at the peak of infection and overall survival, showed no differences between treated and control mice (Fig 3G and 3H).

Since systemic (i.p.) treatment with IL-33 in INF mice did not yield significant effects on the evaluated parameters, we speculated that local administration in peripheral tissue might

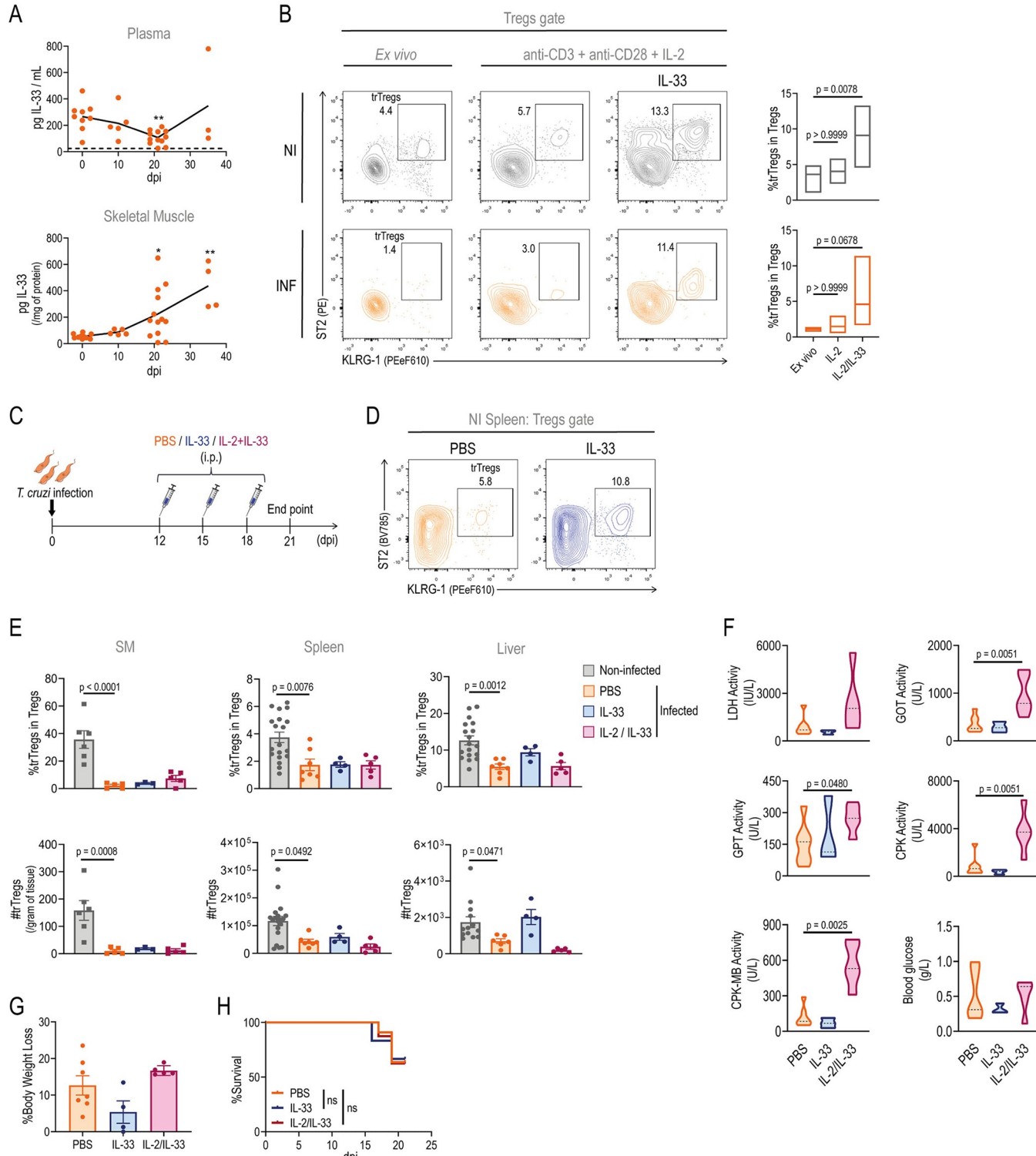

**Fig 3. IL-33 supplementation fails to prevent trTreg reduction in established infection.** (A) IL-33 concentration in plasma and skeletal muscle (SM) lysates obtained from Foxp3-GFP mice at different days post infection (dpi). Muscle IL-33 concentration was normalized to total protein content. Data are presented as individual replicates (circles) and mean (line). Dashed line in plasma indicates the assay's limit of detection. (B) Representative dot plots and cumulative data (N = 4–6 replicates) showing the frequency of ST2+ KLRG-1+ Tregs (trTregs) within total Tregs isolated from the spleen of non-infected (NI) and infected (INF) (21 dpi) Foxp3-GFP mice. Left dot plots correspond to uncultured Tregs (*ex vivo*), while middle and right dot plots correspond to Tregs activated with anti-CD3+anti-CD28+IL-2 with or without IL-33 for 72 hours. Statistical significance was determined by Friedman test. (C) Experimental scheme illustrating

the treatment of Foxp3-GFP mice with an established infection with intraperitoneal (i.p.) IL-33 or IL-33+IL-2. Created with BioRender.com. (D) Dot plots depicting the frequency of trTregs within total Tregs in the spleen of NI Foxp3-GFP mice 72 hours after receiving 3 doses of i.p. IL-33 or PBS, as described in (C). (E-H) Analysis of trTreg response and disease progression in INF mice treated with intraperitoneal IL-33 or IL-33+IL-2 as described in Fig 3C. (E) Graphs displaying trTreg frequency within total Tregs (upper row) and absolute number (bottom row) in SM, liver and spleen at 21 dpi. In all cases, values from untreated NI mice are also shown for comparison. In SM, counts are normalized to tissue weight. (F) Violin plots indicating the distribution of plasma LDH, GOT, GPT, CPK and CPK-MB activities, and glucose concentration at 21 dpi. N = 3–7 per group. (G) Percentage of total body weight reduction at 21 dpi compared to 15 dpi. (H) Survival curve in the different experimental groups. N = 6–11 per group. (E and G) Bars indicate the mean ± SEM. Squares in SM represent values obtained from pools of 2–4 mice, while circles in the remaining plots represent values from individual mice. Statistical significance was determined by Kruskal-Wallis test (A); Friedman test (B); one-way ANOVA (E); Mann-Whitney test (F and G) and Mantel-Cox test (H). P values in (A) are relative to 0 dpi: *p < 0.05; **p < 0.01; while in (E-H) are relative to PBS. Data are representative of two (A) and one (D-H) independent experiments.

better target trTregs. Therefore, we tested the effect of intramuscular (i.m.) IL-33 injections. In this approach, INF mice were injected at 12, 15, and 18 dpi with IL-33 (0.3 μg per muscle) in one hind limb and with PBS in the other as control, following the procedure described by Kuswanto et al., 2016 [10]. Similar to systemic treatment, i.m. IL-33 injection in NI animals resulted in an accumulation of Tregs in SM, with a high proportion of trTregs (S4D Fig). However, in INF animals, the treatment had no effect on Tregs, which were present at low counts, making trTreg evaluation unfeasible (S4E Fig).

Overall, these results indicate that, although trTregs from INF mice are intrinsically able to respond to IL-33, the reduction of trTregs that occurs during acute *T. cruzi* infection cannot be rescued by IL-33 administration, likely due to particular cues generated in the context of an established infection.

## IFN-γ limits trTreg expansion both *in vitro* after IL-33 stimulation and *in vivo* during *T. cruzi* infection

To understand the mechanisms limiting the impact of IL-33 on inducing trTregs during *T. cruzi* infection, we focused on molecules known to counteract IL-33's biological effects, potentially activated by this parasitic infection. Specifically, we evaluated soluble ST2 (sST2), a spliced variant of ST2 lacking the cytosolic and transmembrane domains, which acts as a decoy receptor to neutralize IL-33 activity under various inflammatory conditions [57]. We found undetectable levels of sST2 in the serum from both NI and INF mice (S5A Fig). Within tissues, the quantification of ST2 levels might include both sST2 and the transmembrane isoform (ST2L). Our results showed that ST2 levels were similar between INF mice and NI counterparts in spleen and SM homogenates. In the liver, ST2 levels were remarkably high and exhibited a downregulation during the infection at all dpi tested.

Considering that the maximal trTreg reduction correlates with the highest parasitemia and taking into account previous results indicating that Tregs differentiation is affected by the presence of parasites [48], we further speculated about a possible inhibitory effect from microbial ligands. To evaluate this, we utilized the *in vitro* approach described in Fig 3B, where sorted splenic Tregs from NI mice were cultured for 72 hours in the presence of the trTreg expansion cocktail (anti-CD3, anti-CD28, plus IL-2 and IL-33). For this experiment, heat-killed or lysed trypomastigotes were added during the culture as a source of microbial ligands. Despite the presence of these parasite signals, trTreg expansion in the presence of IL-2 and IL-33 was not impaired (S5B Fig).

Pro-inflammatory cytokines such as IFN-α, IFN-γ, and TNF have been demonstrated to prevent IL-33-mediated expansion of trTregs [49]. Given the induction of multiple inflammatory signals during acute *T. cruzi* infection, as shown in this manuscript (S1B Fig) and previously reported [48,58,59], which may synergize to inhibit IL-33's effect, we explored their potential role in preventing IL-33-mediated trTreg expansion in our experimental setting. To simulate the infection-induced inflammatory context, we used two approaches. First, we

employed transwell cultures designed to emulate the early stages of infection. In this assay, Tregs stimulated with the trTreg expansion cocktail were co-cultured in the upper transwell chamber with splenocytes of NI mice, either alone or in the presence of trypomastigotes. We determined that IL-33 retained its capacity to expand trTregs in these settings (S5C Fig). Second, Tregs stimulated with the trTreg expansion cocktail were cultured in the presence of conditioned media obtained after 24-hour polyclonal stimulation of splenocytes from INF mice at 10 and 21 dpi. In this context, we found that both types of conditioned media appeared to partially reduce IL-33-induced trTreg expansion.

As a final approach to explore potential cytokine candidates present in culture supernatants from INF splenocytes that may restrict IL-33-mediated expansion *in vitro*, we repeated the *in vitro* experiments in the presence of recombinant Th1 cytokines, including IFN-γ, TNF, or IL-12. As shown in the dot plots and cumulative data graphs in Fig 4A, IFN-γ and TNF, but not IL-12, were able to reduce the expansion of trTregs mediated by IL-33 *in vitro*.

Considering the results obtained *in vitro*, we performed an *in vivo* neutralization assay to establish the role of IFN-γ or TNF in the modulation of trTreg dynamics during *T. cruzi* infection (Fig 4B). As expected, given the relevance of these Th1 cytokines for infection control, mice treated with anti-TNF succumbed to the infection (Fig 4C), while mice treated with anti-IFN-γ survived (Fig 4C) but showed an increase in parasitemia (Fig 4D) with no changes in leukocyte numbers in the spleen and a reduction in leukocyte infiltration in SM (Fig 4E). Further evaluation of Tregs dynamics after IFN-γ neutralization showed no changes in the frequency and absolute cell numbers of Tregs in the spleen (Fig 4F) and SM (Fig 4G). A focus on trTregs in the spleen, but not in SM where Tregs numbers were too low for reliable quantification, showed that neutralization of IFN-γ during acute *T. cruzi* infection resulted in an increase in the cell counts of this reparative subset (Fig 4H). While these changes are modest and unlikely to significantly impact on infection progression, they suggest that Th1 cytokines, rather than sST2 or microbial ligands, are relevant modulators of trTreg dynamics during acute *T. cruzi* infection. Furthermore, these findings help to explain the lack of effect observed after IL-33 injection in the established *T. cruzi* infection.

## Early IL-33 administration expands trTregs and other cell subsets, improving disease outcome in infected mice

In a subsequent effort to modulate the trTreg response *in vivo* during acute infection, and considering the results from the previous section indicating that the Th1 responses emerging during infection may limit trTreg expansion, we opted to treat INF mice with IL-33 on 0, 3, and 6 dpi (Fig 5A). As initial indicators of the treatment's impact, we observed that IL-33 significantly improved survival rates (Fig 5B) and reduced weight loss during the acute phase (S6A Fig), suggesting a protective effect on overall disease progression. In agreement, the evaluation of biochemical markers at 21 dpi, a time point characterized by marked tissue damage, showed that IL-33 treatment ameliorated the hypoglycemia resulting from infection (Fig 5C). Moreover, early IL-33 supplementation reduced LDH, GOT, and CPK activity, while it did not modify GPT activity and led to a tendency for decreased CPK-MB activity. Finally, we compared the degree of SM damage in IL-33-treated versus control mice (Fig 5D and S6B Fig). Histological examination of the control group showed severe necrosis and moderate to severe dystrophic calcifications in ~70% of the samples, accompanied by predominantly multifocal mononuclear inflammatory infiltrates in ~85% of samples. In contrast, IL-33 administration improved histological parameters, with severe or moderate necrosis and dystrophic calcification reduced to ~30% of samples, with no changes in infiltration. Thus, our analysis revealed

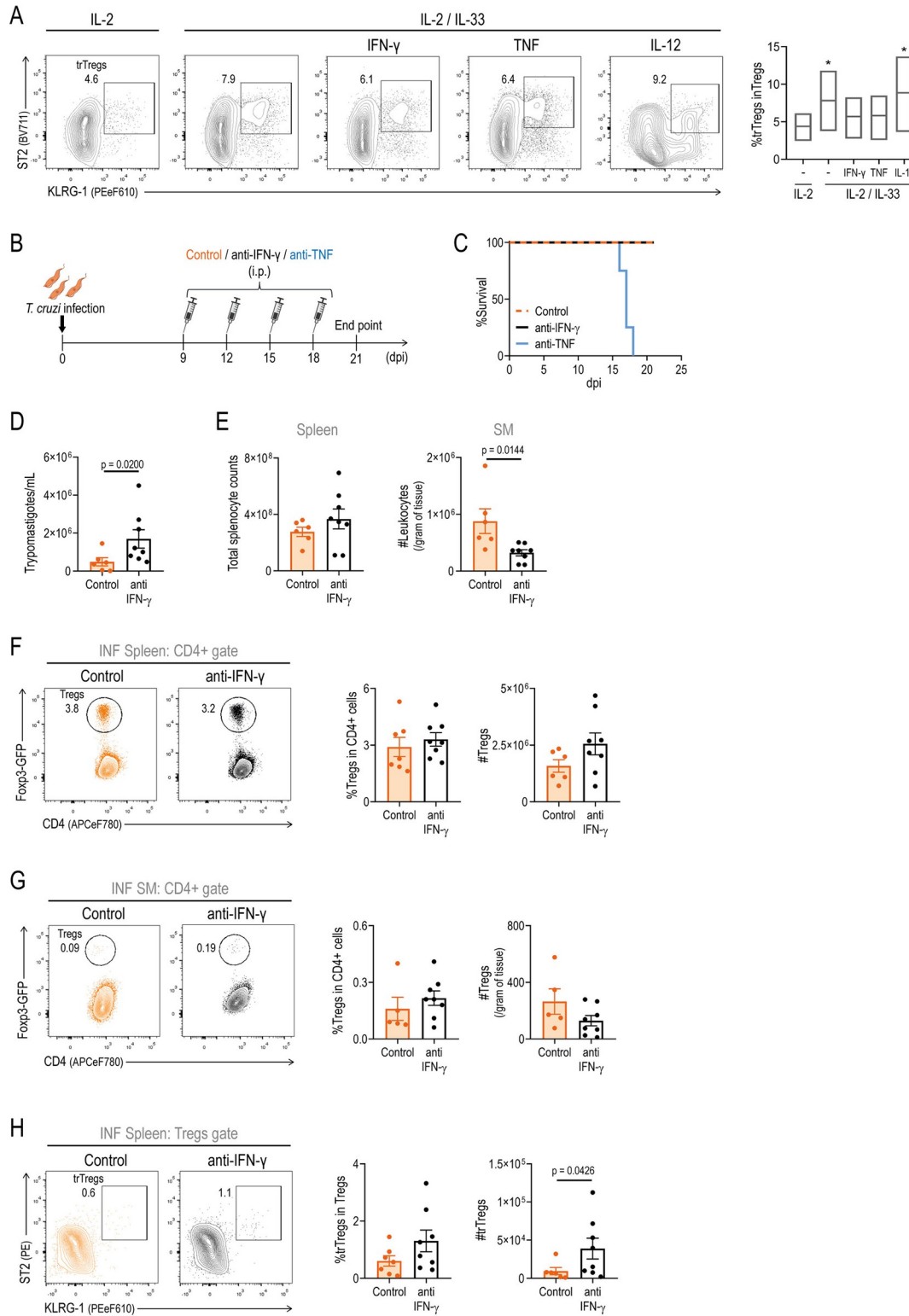

**Fig 4. IFN-γ limits trTreg expansion both *in vitro* after IL-33 stimulation and *in vivo* during *T. cruzi* infection.** (A) Representative dot plots and cumulative data (N = 3) showing the frequency of ST2+ KLRG-1+ Tregs (trTregs) within total Tregs isolated from the spleen of non-infected (NI) Foxp3-GFP mice and incubated for 72 h with anti-CD3+anti-CD28 together with the addition of IL-2; IL-2+IL-33 and IL2+IL33 plus cytokines associated to Th1 signals. Statistical significance was determined by Friedman test. P values are relative to IL-2 condition: *p < 0.05. (B) Experimental scheme illustrating the

treatment of Foxp3-GFP mice with anti-IFN-γ or anti-TNF at different days post infection (dpi). Created with BioRender.com. (C) Survival curve in the different experimental groups. N = 4–8 per group. (D-H) Analysis at 21 dpi of mice receiving anti-IFN-γ or isotype control. (D) Parasite counts in blood. (E) Total splenocyte counts (left) and skeletal muscle (SM) leukocyte counts (right). (F-H) Representative dot plots showing Tregs frequency within CD4+ cells in spleen (F) and SM (G), and trTreg frequency within total Tregs in spleen (H). Bars on the right indicate the frequency and absolute numbers (mean ± SEM) of the gated cell populations in the respective dot plots on the left. (E-H) Circles in bars represent individual replicates. (E and G) Cell counts in SM are normalized to tissue weight. Statistical significance was determined as follow: Mantel-Cox test (C) Mann-Whitney test for parasitemia (D and G) and unpaired t test (E, F and H). Data were collected from 1 (anti-TNF treatment) or 2 (anti-IFN-γ or isotype control treatment) experiments.

that IL-33 treatment reduced tissue damage in SM, as evidenced by milder necrosis and calcification compared to control mice.

Next, we sought to understand the mechanisms underlying the improved disease progression and reduced SM damage by first evaluating Tregs and trTreg dynamics. Early IL-33 treatment had an inconsistent effect on the total Treg pool, with conserved frequencies in SM, liver, and spleen, and increased Treg cell counts only in the spleen (S6C Fig). In contrast, IL-33 supplementation consistently expanded trTregs across all the evaluated tissues. Specifically, we observed a tendency for increased frequency and significantly higher absolute numbers of trTregs in the SM of IL-33-treated INF mice, along with significant increases in both parameters in the liver and the spleen (Fig 5E and 5F).

Given that different immune cell types express ST2 and may respond to IL-33 [20,57], we evaluated whether IL-33 treatment during *T. cruzi* infection affected other ST2-expressing cell subsets. Among these, we particularly assessed type 2 Innate lymphoid cells (ILC2) as they are also involved in tissue damage control [52,60]. Using a gating strategy adapted from Tait Wojno and Beamer [61] (S6D Fig), we examined ILC2 infiltration in various tissues and found that both ILC2 frequency and absolute numbers were increased in SM, liver and spleen of INF mice after early IL-33 injection (Fig 5G and 5H). These findings underscore the effectiveness of early IL-33 treatment in sustaining elevated trTregs and ILC2 levels over a two-week period, up to the peak of infection.

In addition to trTregs and ILC2, IL-33 can activate effector immune cells either directly through ST2 signaling on the target cell or indirectly by inducing the production of intermediate cues, such as IFN-γ or TNF [57]. Therefore, we evaluated whether early IL-33 administration affected parasite specific CD8+ T cells, an effector response critical for *T. cruzi* control [62]. Interestingly, IL-33-treated mice exhibited increased frequencies of this cell subset in SM, liver, as well as in spleen (Fig 5I). These frequency changes corresponded with conserved counts of parasite-specific CD8+ T cells in SM and liver, and increased counts in the spleen (Fig 5J). In line with these findings, INF mice that received IL-33 injections showed significantly decreased parasitism in all the evaluated tissues, including SM, liver and spleen (Fig 5K). Given the enhanced antiparasitic CD8+ T cell response and the relevance of IFN-γ or TNF for its development, we evaluated whether early IL-33 modulated systemic levels of these inflammatory signals. Notably, we found no differences in plasma IFN-γ concentration between experimental groups, while TNF levels were reduced by IL-33 administration (Fig 5L). These results indicate that, despite conserved or reduced levels of classic Th1 soluble mediators, early IL-33 treatment can potentiate antiparasitic immune response, likely as a result of increased effector parasite-specific CD8+ T cell responses.

To further delineate how IL-33 influences different immune cell subsets and contributes to the improved infection outcome, we assessed immunological and disease-related parameters at 10 dpi, a critical time point marked by the emergence of the adaptive antiparasitic response and the onset of tissue damage. At this earlier stage of infection, we found a distinct response, evidencing particular kinetics for IL-33's effects. For instance, in trTregs, we observed a

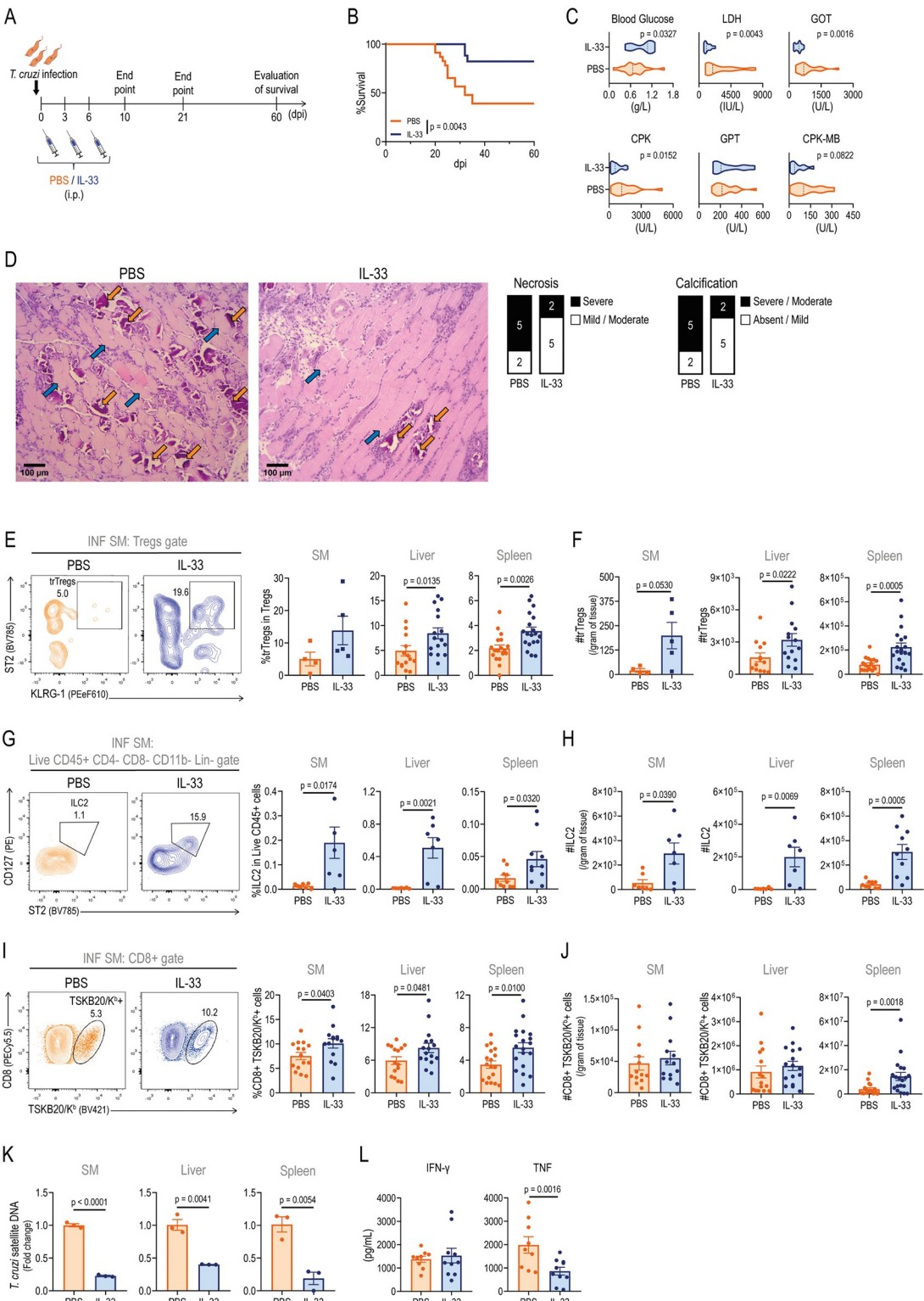

**Fig 5. Early IL-33 administration expands trTregs and improves disease outcome in infected mice.** Immune response and disease progression was evaluated in infected (INF) Foxp3-GFP mice after receiving intraperitoneal (i.p.) IL-33 or PBS the day of infection and on 3 and 6 days post infection (dpi). (A) Experimental scheme. Created with BioRender.com. (B) Survival curve. N = 17–23 per group. (C-L) Evaluation of treated INF mice at 21 dpi. (C) Violin plots indicating the distribution of plasma glucose concentration and LDH, GOT, CPK, GPT and CPK-MB activities. N = 9–18 per group. (D) Representative Hematoxylin-

Eosin staining of quadriceps muscle from both groups. N = 7 per group. Blue arrows indicate necrotic muscle fibers; orange arrows indicate dystrophic calcification. Magnification: 10X. Bars on the right display the proportion of samples with varying levels of muscle fiber necrosis or calcification in each group. (E) Representative dot plots showing the frequency of ST2+ KLRG-1+ Tregs (trTregs) within total Tregs isolated from the skeletal muscle (SM). Bars displaying trTreg frequency within total Tregs in SM, liver and spleen. (F) Bars indicating trTreg count in SM, liver and spleen. (G) Representative dot plots illustrating SM type 2 innate lymphoid cells (ILC2) frequency within CD45+ CD4- CD8- Lin (CD3, CD19, NK1.1, CD11c)- CD11b- cells. Bars displaying ILC2 frequency within Live CD45+ cells in SM, liver and spleen. (H) ILC2 count in SM, liver and spleen. (I) Representative dot plots showing TSKB20/K$^b$ staining in SM CD8+ cells. Bars indicating frequency of parasite-specific CD8+ T cells in SM, liver and spleen. (J) Parasite-specific CD8+ cell count in SM, liver and spleen. (K) *T. cruzi* satellite DNA quantification in SM, liver and spleen. For each tissue, the graphs represent the fold change in parasitic load in IL-33 treated animals relative to the PBS-treated group. (L) Plasma IFN-γ and TNF concentration. (E-L) Bars represent the mean ± SEM and circles represent individual replicates. For SM, squares represent values obtained from pools of 2–4 mice (E and F) and counts are normalized to tissue weight (F, H and J). Statistical significance was determined as follow: Mantel-Cox test (B), unpaired t test (E-I, K and L) and Mann-Whitney test (J). (C) Unpaired t test was used for CPK-MB activity while Mann-Whitney test was applied to the remaining parameters. Data are representative of two (B-D, G and H), three (E, F and L) and four (C, I, J and K) independent experiments.

tendency for increased frequency and absolute numbers in non-lymphoid target tissues such as SM (S7A and S7B Fig). In the spleen, where trTreg commitment occurs, we observed a significantly higher frequency, alongside a tendency for augmented counts. Similarly, ILC2 showed a tendency toward higher frequencies and absolute numbers in SM and spleen of IL-33 treated mice (S7C and S7D Fig). Notably, parasite-specific CD8+ T cell frequencies and counts remained stable in both tissues (S7E and S7F Fig).

Finally, we assessed markers of infection progression, including biochemical indicators of tissue damage, tissue parasitism, and effector Th1 cytokines. In line with the trTregs and ILC2 responses, early infection in IL-33-treated mice progressed with a tendency toward reduced GOT and CPK levels, with no changes in LDH and GPT activity, or glucose concentrations (S7G Fig). Additionally, IL-33 administration led to a mild increase in parasite burden in SM at 10 dpi, while no changes were observed in the spleen (S7H Fig). Plasma concentrations of IFN-γ and TNF were comparable between experimental groups at this time point (S7I Fig).

Altogether, these findings demonstrate that early IL-33 administration can improve the course of acute infection by reducing tissue damage and enhancing parasite control. This effect is likely due to the combined action of IL-33 on various immune cell subsets, including trTregs, ILC2, and effector CD8+ T cells. The results obtained at 10 dpi particularly highlight the importance of reparative populations, such as trTregs and ILC2 in mitigating tissue damage during *T. cruzi* infection, even in the absence of significant changes in CD8+ T cell response and parasitism at this early stage.

## Discussion

Previous reports indicate that tissue injury, whether sterile or infection-derived, is associated with increased IL-33 release following cell destruction [20], which contributes to Tregs accumulation. The increased presence of Tregs, in turn, sustains reparative mechanisms, as has been well described in sterile injury context, particularly within SM [4,9,10]. In the context of murine acute *T. cruzi* infection, a model for Chagas' disease with a profound compromise of various target tissues, including SM, our results reveal a novel scenario marked by a diminished trTreg response and reduced plasmatic IL-33 concentration despite extensive tissue damage. This suggests a specific impairment in the regulatory fate associated with tissue repair, complementing our earlier work which reported limited Tregs responses but with the acquisition of a type 1 specialization profile during *T. cruzi* infection [48,63]. Altogether, our findings delineate a scenario where the reduction of a reparative/regulatory cell subset during the acute phase may potentially facilitate tissue damage by compromising physiological repair

mechanisms. Consistent with this, a detailed characterization of tissue damage associated with *T. cruzi* infection, particularly in SM, revealed a clear modification in the SM transcriptome with activation of immune-related pathways and a general downregulation of genes associated with myogenesis, adipogenesis, and oxidative phosphorylation pathways, linked to normal muscle physiology and repair processes after injury [64–66]. Therefore, our results provide evidence that during acute *T. cruzi* infection, the transcriptional landscape in SM denotes damage and correlates with a reduced frequency of trTregs. This association may potentially connect deficient repair processes with long-term consequences in chronic immunopathology during Chagas' disease. Further studies may be required to deeply assess these possible connections.

The attenuation of the reparative response during acute *T. cruzi* infection was systemic, manifested by reduced frequency and absolute numbers of trTregs across evaluated tissues, including SM, liver, and spleen. Unexpectedly, IL-33 levels exhibited both consistency and discrepancy: plasma IL-33 concentration diminished, correlating with the limited trTreg responses, while muscle IL-33 levels increased, yet failed to prevent the reduction of trTregs in that tissue. Two aspects of these findings were puzzling: the unexpected reduction in systemic IL-33 levels despite documented tissue damage in *T. cruzi* infection and the lack of an accompanying increase in trTregs in muscle despite an elevated local IL-33 concentration. The mechanisms regulating IL-33, which are complex and varied, include its rapid degradation after tissue injury [20,67,68]. For example, caspases 1 and 7, produced in inflammatory contexts and involved in the death of infected cells, fragment IL-33 to inactivate it [69]. Additionally, IL-33 is easily oxidizable in the extracellular space, limiting ST2-dependent immunological responses [70]. Accordingly, the inflammatory milieu during acute *T. cruzi* infection could rapidly oxidize and/or degrade IL-33, diminishing its availability, especially in plasma. In contrast, elevated IL-33 levels were detected in muscle. However, this quantification was conducted in tissue lysates, presenting technical limitations that impede confirmation of the extracellular presence and, consequently, the availability of IL-33 for ST2+ cells. Discrepancies between IL-33 concentration and trTreg numbers in SM suggest potential unavailability or counteraction by inflammatory signals. Alternatively, trTregs cells may face high mortality or reduced generation during the acute stage. Considering the multi-step development of trTregs beginning in the spleen [19,71], the reduction in IL-33 levels in this organ during acute infection could be linked to a lower generation of these cells and, consequently, their reduced arrival at target organs.

Given the scenario described earlier, and our aim to understand the impact of trTreg reduction on disease progression, we intended to boost the numbers of this cell subset through IL-33 supplementation in established *T. cruzi* infection. However, IL-33 treatment, administered systemically or locally around the second week of infection, failed to rescue trTreg numbers and had no effect on disease progression. Notably, the lack of response to IL-33 in terms of trTreg expansion was specific to the infection condition, as this cytokine increased trTregs in non-infected animals. Remarkably, our *in vitro* approaches, while operating under high stimulatory conditions—a limitation to consider when extrapolating these results—provided a proof of principle demonstrating that trTregs from infected mice retain intrinsic responsiveness to IL-33. These findings suggest that the restrictive *in vivo* environment imposed by infection hinders IL-33-induced trTreg expansion. Indeed, we showed that IFN-γ, one of the hallmark effector cytokines involved in *T. cruzi* control, plays a role in limiting trTreg frequency and cell counts during the acute phase of this parasite infection. These results are in agreement with a previous report demonstrating that IFN-α, IFN-γ, and TNF inhibited IL-33-induced expansion of the VAT-Treg population and that these cytokines may induce trTreg loss by direct and indirect mechanisms during inflammatory conditions such as obesity [72]. Altogether, this report and our results support a scenario where a strong Th1 inflammatory

environment may restrict the acquisition of a trTreg fate while favoring other specialization pathways, such as the acquisition of Th1 characteristics, as previously reported for Tregs by our group [48,63].

To explore the effects of IL-33 while avoiding the restrictive environment present at the peak of *T. cruzi* infection, we implemented early treatment during the first week of infection. This approach resulted in a less severe acute infection, with increased survival, reduced tissue damage, and improved parasite control. In line with previous studies [10,24,28], early IL-33 supplementation led to a significant and consistent expansion of trTregs not only in secondary lymphoid organs, such as the spleen, but also in *T. cruzi* target organs like the liver and SM. By contrast, IL-33 treatment had minimal impact on the total Tregs pool, with increases observed only in splenic Tregs counts. Notably, the accumulation of trTregs, particularly in SM, correlated with reduced local damage, including less necrosis and calcification, as determined by histological evaluation.

Given IL-33's ability to modulate various immune cell subsets [67], we also assessed its effects on ILC2 and parasite-specific CD8+ T cells, both of which were also expanded by IL-33 treatment. ILC2 play crucial roles in combating certain infectious agents and, like trTregs, promote tissue repair processes [60]. Indeed, IL-33-mediated ILC2 expansion has been linked to resistance against cerebral malaria and various intestinal infections [27,73–75]. Our findings that IL-33 enhanced antiparasitic CD8+ T cell immunity align with previous research suggesting a role for this alarmin in inducing robust antiviral responses [76,77]. Furthermore, IL-33 has been shown to preserve CD8+ T cell stemness and re-expansion capacity in chronic viral infections. However, our results do not necessarily indicate a direct effect of IL-33 on this effector subset.

Evaluations at different time points post-infection revealed that IL-33's effects on various immune cell subsets followed distinct kinetics. TrTregs and ILC2 appeared to respond directly to IL-33, with increased numbers evident as early as 10 dpi. This increase was particularly notable for trTregs in secondary lymphoid organs, where their commitment to this specialized fate occurs. This early increase, which was evident before the initiation of parasite-specific responses and the onset of acute damage, was further amplified over the course of the infection, leading to significantly higher numbers at the infection's peak in various tissues. In contrast, the augmented parasite-specific CD8+ T cell responses were not observed in the early stages of infection, suggesting that IL-33's effect on this population might be indirect, likely mediated through other IL-33-responsive cell subsets. Indeed, it has been reported that the IL-33/ILC2 axis can potentiate CD8+ T cell recruitment, activation, and proliferation in cancer settings [78–80].

Interestingly, the simultaneous expansion of trTregs and parasite-specific CD8+ T cells might seem counterintuitive, given the well-established association of regulatory responses with diminished antimicrobial immunity [81]. However, these findings can be reconciled by considering the distinct roles of specialized Treg subsets. Our previous studies in the context of *T. cruzi* infection demonstrated that systemic depletion of total Tregs in infected DEREG mice enhanced CD8+ T cell immunity and parasite control, likely due to the suppression exerted by Th1-like Tregs on effector CD8+ T cells through CD39-mediated mechanisms [48,63]. In contrast, the current study highlights the role of trTregs, a specialized subset with a distinct program that, when expanded by IL-33 during early *T. cruzi* infection, may promote tissue repair without significantly suppressing CD8+ T cell responses. While Tregs are traditionally associated with limiting anti-pathogen immunity, certain Tregs have also been shown to support the recruitment and maintenance of protective CD8+ T cells in tissues under specific conditions [82–84]. Moreover, tissue-resident Tregs can sustain local CD8+ T cell responses critical for controlling the protozoan parasite *Eimeria vermiformis* [85]. Thus,

trTregs may exhibit negligible suppressive potential on CD8+ T cells or could even contribute to enhancing CD8+ T cell immunity in the context of IL-33 supplementation.

In summary, our findings underscore the positive impact of IL-33 supplementation during early *T. cruzi* infection, contributing to reduced tissue damage and improved parasite control. Building upon existing research, we propose that this beneficial effect cannot be solely attributed to trTreg modulation. Rather, it likely arises from the concerted action of expanding cell subsets involved in tissue repair, such as trTregs and ILC2, alongside those engaged in microbial control, particularly CD8+ T cells. Future studies are needed to meticulously dissect the crosstalk and specific roles of each immune cell subset in mediating the effects of IL-33 on *T. cruzi* infection outcomes. It remains to be determined how these effects during the acute phase influence the progression of chronic disease or whether IL-33 treatment in the chronic phase itself could impact clinical outcomes. To date, only one study has evaluated IL-33 expression in chronic Chagas disease patients, finding no correlation with disease severity [86]. Therapies involving IL-33 could be developed to modulate the immune system, favoring a specific combination of regulatory and effector responses. The goal would be to achieve effective pathogen clearance while minimizing collateral damage, thereby preventing clinical pathology.

## Materials and methods

### Ethics statement

Mouse handling followed international ethical guidelines. All experimental procedures were conducted in compliance with the ethical standards set by the Institutional Animal Care and Use Committee of Facultad de Ciencias Químicas–Universidad Nacional de Córdoba, and were approved under protocol number RD-733/2018 and RD 2134/2022.

### Mice

Age-matched (8 to 12 week-old) mice of both sexes were used. Foxp3-GFP reporter mice (B6. Cg-Foxp3tm2Tch/J—RRID:IMSR_JAX:006772) were obtained from The Jackson Laboratories (USA). BALB/c mice were obtained from School of Veterinary, La Plata National University (La Plata, Argentina). Animals were bred in the animal facility of the Facultad de Ciencias Químicas, Universidad Nacional de Córdoba, and housed under a 12:12 h light-dark cycle with food and water ad libitum. The institutional animal facility follows the recommendations of the Guide for the Care and Use of Experimental Animals, published by the Canadian Council for the Protection of Animals.

### Parasites and experimental infection

For all experiments, *T. cruzi* Tulahuen strain was used. Bloodstream trypomastigotes were maintained in male BALB/c mice by serial passages every 10–11 days. For *in vivo* assays, Foxp3-GFP reporter mice were inoculated intraperitoneally with 0.2 mL PBS containing 5,000 trypomastigotes.

For *in vitro* assays, trypomastigotes were obtained from the extracellular medium of infected monolayers of Vero cells cultured in RPMI 1640 medium (Gibco, Cat# 21870–100) containing 10% heat inactivated fetal bovine serum (FBS, Natocor), 2 mM glutamine (Gibco, Cat# 35050061), 10 mM HEPES (Gibco, Cat# 15630080) and 40 ug/mL gentamicin (Veinfar Laboratories). After 7 days of infection, extracellular medium was collected, centrifuged at 1800 g for 30 min at room temperature and incubated for 2 h at 37˚C. Trypomastigotes were recovered from the supernatant and counted using a Neubauer chamber. Heat-killed trypomastigotes were obtained after incubation at 56˚C for 10 min (adapted from [87]), while lysed

trypomastigotes were obtained after 3 cycles of freeze/thaw and 5 minutes of sonication (adapted from [88]).

## Parasite quantification in blood and tissues

Parasitemia was monitored by counting the number of viable trypomastigotes in blood after lysis with a 0.87% ammonium chloride buffer. For tissue parasite quantification, genomic DNA was purified from 50 μg of tissue (spleen, liver, SM and heart) using TRIzol Reagent (Life Technologies, Cat# 15596026) following manufacturer´s instructions. Satellite DNA from *T. cruzi* (Taqman, Cat# 4332078, ID: AP47V73) was quantified by real time PCR using TaqMan Universal Master Mix II, no UNG (Applied Biosystem, Cat# 4440047) using the primer and probe sequences described by Piron et al. [89]. The samples were subjected to 45 PCR cycles in a thermocycler StepOnePlus Real-Time PCR System (Applied Biosystems, RRID:SCR_015805). Abundance of satellite DNA from *T. cruzi* was normalized to the abundance of GAPDH (Taqman Rodent GAPDH Control Reagent, Applied Biosystem, Cat# 4352339E), quantified through the comparative CT method and expressed as arbitrary units, as previously reported [48,50,90].

## Cell preparation

To obtain cell suspensions from solid tissues, euthanized mice were perfused with 10 mL cold Hanks' Balanced Salt Solution (Gibco, Cat# 14185052). Spleens and livers were obtained and mashed through a tissue strainer. Liver infiltrating cells were obtained after 25 min centrifugation (600 g without brake) in a 35% and 67.5% bilayer Percoll (GE Healthcare, Cat# 17-0891-01) gradient. The interphase containing leukocytes was recovered and washed. Erythrocytes in spleen and liver cell suspensions were lysed for 3 min in ACK Lysing Buffer (Gibco, Cat# A10492-01). Heart and skeletal muscle (quadriceps, gastrocnemius and tibialis anterior) were excised, minced and digested for 30 min in collagenase D (2 mg/mL, Roche, Cat# COLLD-RO) and DNase I (100 μg/mL, Roche, Cat# 04536282001). Digested tissues were filtered through a 70 μm filter and washed. Infiltrating leucocytes were obtained after 25 min centrifugation (600 g without brake) in a 40% and 75% bilayer Percoll gradient. The interphase was recovered and washed. Cell numbers were counted in Turk's solution using a Neubauer chamber. For experiments evaluating trTregs in SM, cell suspensions from pools of 2–4 mice were used when indicated in the figure legends due to the low frequency of this subset.

## *In vitro* assays

For Tregs and Tconv culture, cells were purified from NI or 21 dpi Foxp3-GFP mice. CD4+ cells were isolated from pooled splenic suspensions by magnetic negative selection using EasySep Mouse CD4+ T Cell Isolation Kit (StemCell Technologies, Cat# 19852) according to manufacturer's instruction. Afterwards, the enriched CD4+ T cell suspension was surface stained and Tregs and Tconv were further purified by cell sorting with a BD FACSAria II Cell Sorter (RRID:SCR_018934) according to the following phenotype: Tregs (CD4+ Foxp3-GFP+) and Tconv (CD4+ Foxp3-GFP-). Purified cells (75000 cells/well) were cultured for 3 days in 96-well U bottom plates coated with 2 μg/mL anti-CD3 (Thermo Fisher Scientific Cat# 14-0031-85, RRID:AB_467050) and 1 μg/mL anti-CD28 (Thermo Fisher Scientific Cat# 14-0281-86, RRID:AB_467192) supplemented with 10 ng/mL of recombinant mIL-2 (Biolegend, Cat# 714604) to allow Tregs survival. Cells were cultured in complete culture media containing RPMI 1640 medium (Gibco, Cat# 21870–100) 10% heat inactivated FBS (Natocor), 2 mM glutamine (Gibco, Cat# 35050061), 55 μM 2-mercaptoethanol (Gibco, Cat# 21985023) and 80μg/mL gentamicin (Veinfar Laboratories). When indicated, media contained 50 ng/mL

recombinant mIL-33 (R and D, Cat# 3626-ML/CF) alone or combined with the following murine recombinant cytokines: IFN-γ (50 ng/mL, Immunotools, Cat# 12343537), TNF (50 ng/mL, Immunotools, Cat# 12343010) and IL-12p70 (10 ng/mL, Peprotech, Cat# 210–12). Alternatively, 50 μL of conditioned media was used.

In co-culture transwell experiments, Tregs from NI mice were placed in the bottom of the culture plate in the presence of recombinant mIL-33 and splenocytes (1:2.5 ratio) alone or with trypomastigotes (1:10 ratio) that were placed in the transwell chamber.

In co-cultures with parasite ligands, Tregs from NI animals were incubated with heat-killed or lysed trypomastigotes (ratio 1:1) in the presence of recombinant mIL-33.

When indicated, live cell numbers after culture were counted in 0.2% trypan's blue solution (Gibco, Cat# 15250061) using a Neubauer chamber.

## Conditioned media generation

For conditioned media, total splenocytes were isolated from pooled splenic suspensions of 10 dpi and 21 dpi Foxp3-GFP mice. Cell suspensions ($5 \times 10^6$ cells/mL) were cultured for 24 h in 24-well plates in complete culture media supplemented with 50 ng/mL PMA (Sigma-Aldrich, Cat# P1585) and 1 μg/mL ionomycin (Sigma-Aldrich, Cat# I0634).

## Biochemical determinations

Plasma was collected after blood centrifugation for 8 min at 3000rpm. Quantification of biochemical markers of tissue damage was performed at Laboratorio Biocon (Córdoba, Argentina) using a Dimension RXL Siemens analyzer. GOT, GPT, LDH and CPK activity was determined by UV kinetic method, CPK-MB activity was evaluated by enzymatic method, while glucose concentration was assessed by kinetic/colorimetric method.

## IL-33, ST2, IFN-γ and TNF quantification

IL-33 concentration was determined with an IL-33 Mouse ELISA kit (Invitrogen, Cat# 88–7333, limit of detection: 25 pg/mL), while ST2 was quantified using a Mouse ST2/IL-33R Duo-Set ELISA kit (R and D Systems, Cat# DY1004, limit of detection: 312.5 pg/mL) in plasma and tissue lysates. Plasma samples were obtained as previously described. Tissue lysates were obtained after centrifugation at 10000 g during 10 min of tissue samples homogenized in PBS containing 0,5% BSA, 0,4 M NaCl, 1 mM EDTA, 0,05% Tween 20 and a protease inhibitor cocktail (Roche, Cat# COEDTAF-RO) (adapted from [91]). GraphPad Prism software (RRID: SCR_002798) version 8.0.1 was used to generate the calibration curve and determine IL-33 and ST2 concentration. In tissue lysates, values were normalized to total protein content determined using Bradford reagent (BioRad, Cat# 5000006). Two Synergy HT Multi-mode microplate reader (Biotek, RRID:SCR_020536) was used to determine absorbances at 450 nm (ELISA) and 595 nm (protein quantification).

IFN-γ and TNF levels in plasma were determined using LEGENDplexTM Multi-Analyte Flow Assay Kit (Biolegend, limit of detection: 78.1 pg/mL for both cytokines) for Mouse Th Cytokine Panel according to manufacturer's instructions.

## Flow cytometry

Combinations of the following antibodies were used for flow cytometry: biotin polyclonal anti-Amphiregulin (R and D Systems Cat# BAF989, RRID:AB_2060662), PE anti-BATF clone S39-1060 (BD Biosciences Cat# 564503, RRID:AB_2738829), Super Bright 645 anti-CD11b clone M1/70 (Thermo Fisher Scientific Cat# 64-0112-80, RRID:AB_2662386), PE-Cyanine7

anti-CD11c clone N418 (Thermo Fisher Scientific Cat# 25-0114-82, RRID:AB_469590), PE anti CD127 clone A7R34 (Thermo Fisher Scientific Cat# 12-1271-83, RRID:AB_465845), PE-Cyanine7 anti-CD19 clone eBio1D3 (Thermo Fisher Scientific Cat# 25-0193-82, RRID: AB_657663), PE-Cyanine7 anti-CD3 clone 145-2C11 (Thermo Fisher Scientific Cat# 25-0031-82, RRID:AB_469572), APC anti-CD4 clone GK1.5 (BioLegend Cat# 100412, RRID: AB_312697), Super Bright 645 and APC-eFluor 780 anti-CD4 clone GK1.5 (Thermo Fisher Scientific Cat# 64-0041-82, RRID:AB_2717079 and Cat# 47-0041-82, RRID:AB_11218896 respectively), PE-Cyanine5 anti-CD44 clone IM7 (Thermo Fisher Scientific Cat# 15-0441-81, RRID:AB_468748), Alexa Fluor 700 anti-CD45 clone 30-F11 (Thermo Fisher Scientific Cat# 56-0451-82, RRID:AB_891454), APC-Cyanine7 anti-CD45 clone 30-F11 (BD Biosciences Cat# 557659, RRID:AB_396774), APC-eFluor 780 anti-CD62L clone MEL-14 (Thermo Fisher Scientific Cat# 47-0621-82, RRID:AB_1603256), PE-Cyanine5.5 anti-CD8 clone 53–6.7 (Thermo Fisher Scientific Cat# 35-0081-82, RRID:AB_11217674), Brilliant Violet 605 anti CTLA-4 clone UC10-4B9 (BioLegend Cat# 106323, RRID:AB_2566467), FITC anti-Foxp3 clone FJK-16s (Thermo Fisher Scientific Cat# 11-5773-82, RRID:AB_465243), Super Bright 600 anti-GITR clone DTA-1 (Thermo Fisher Scientific Cat# 63-5874-80, RRID:AB_2688054), PerCp-eFluor 710 anti-ICOS clone 7E.17G9 (Thermo Fisher Scientific Cat# 46-9942-82, RRID: AB_2744728), PerCp-eFluor 710 anti-IRF4 clone 3E4 (Thermo Fisher Scientific Cat# 46-9858-82, RRID:AB_2573912), eFluor 660 anti-Ki-67 clone SolA15 (Thermo Fisher Scientific Cat# 50-5698-82, RRID:AB_2574235), PE-eFluor 610 anti-KLRG-1 clone 2F1 (Thermo Fisher Scientific Cat# 61-5893-82, RRID:AB_2574630), PE-Cyanine7 anti-NK1.1 clone PK136 (BioLegend Cat# 108714, RRID:AB_389364), Brilliant Violet 421 anti-PD-1 clone 29F.1A12 (BioLegend Cat# 135221, RRID:AB_2562568), PE and Brilliant Violet 785 anti-ST2 clone DIH9 (BioLegend Cat# 145304, RRID:AB_2561915 and Cat# 145321, RRID:AB_2860702 respectively) and PerCp-eFluor 710 anti-TIGIT clone GIGD7 (Thermo Fisher Scientific Cat# 46-9501-82, RRID:AB_11150967). To detect biotinylated antibodies, Streptavidin Qdot 605 (Invitrogen) was used.

For surface staining, cell suspensions were incubated with fluorochrome labeled-antibodies together with LIVE/DEAD Fixable Aqua Dead Cell Stain Kit, for 405 nm excitation (Invitrogen, Cat# L34966) in PBS 2% FBS for 20 min at 4°C. To identify *T. cruzi* specific CD8+ T cells, cell suspensions were incubated with an H-2Kb *T. cruzi* trans-sialidase amino acids 569–576 ANYKFTLV (TSKB20) APC- or Brilliant Violet 421- labeled Tetramer (NIH Tetramer Core Facility) for 20 min at 4°C, in addition to the surface staining antibodies.

For transcription factors detection, cells were initially stained for surface markers, washed, fixed, permeabilized and stained with Foxp3/Transcription Factor Staining Buffers (eBioscience, Cat# 00-5523-00) according to eBioscience One-step protocol for intracellular (nuclear) proteins. For intracellular cytokine detection, $2 \times 10^6$ cells per well were cultured in 200 μL supplemented RPMI 1640 medium and stimulated during 2 h at 37°C with 50 ng/mL PMA (Sigma-Aldrich, Cat# P1585) and 1 μg/mL ionomycin (Sigma-Aldrich, Cat# I0634) in the presence of Brefeldin A and Monensin (eBioscience, Cat# 00-4506-51 and Cat# 00-4505-51 respectively). Then, stimulated cells were surface-stained as indicated above, fixed and permeabilized with Intracellular Fixation & Permeabilization Buffer Set (eBioscience) or IC Fixation Buffer and permeabilization Buffer (BD bioscience) following manufacturers' indications. In all cases, intracellular staining was performed by a 30 min incubation at room temperature.

All samples were acquired on FACSCanto II (BD Biosciences, RRID:SCR_018056), LSRFortessa (BD Biosciences, RRID:SCR_018655) or Attune-NxT (Life Technologies, RRID: SCR_019590) and data were analyzed with FlowJo software (RRID:SCR_008520) version X.0.7.

## RNA sequencing

Perfused mouse quadriceps were obtained and stored in RNAlater Stabilization Solution (Life Technologies, Cat# AM7020) at -80˚C. Then, 25 mg of tissue was dissociated using Bead Ruptor Elite (Omni International) for 45 seconds at 4.85 m/s and RNA was isolated using RNeasy Fibrous Tissue Mini Kit 50 (Qiagen, Cat# 74704) following manufacturer's indications. RNA concentration was determined with a Qubit 2.0 Fluorometer (Thermofisher, RRID:SCR_020553), while a 2100 Bioanalyzer instrument (Agilent, RRID:SCR_018043) was used for quality evaluation. Poly (A) mRNA Magnetic Isolation Module (New England BioLabs, Cat# E7490) was used for cDNA library preparation according to manufacturer's protocol. Quality control of libraries was determined as described for RNA. Quantification of cDNA libraries was performed with PerfeCTa NGS Quantification Kit for Illumina Sequencing Platforms (QuantaBio, Cat# 95154–500). For each experimental group, 3 biological replicates were sequenced with NextSeq 550 System (Illumina, RRID:SCR_016381). For data analysis, a Salmon index was built from Gencode [92] Mouse release M27 (GRCm39) [GENCODE - Mouse Release M27 (gencodegenes.org)] using Salmon [93] v1.5.1 [Release Salmon 1.5.1 · COMBINE-lab/salmon · GitHub] with default k-mer size (31) and the—gencode flag. FASTQ sequence reads (SRA accession PRJNA941341) were mapped to the M27 index and transcript abundances were estimated using salmon quant on 8 threads. Salmon quant files were subsequently loaded into R v4.1.0 using tximeta v1.10.0, and differentially expressed genes were called using default parameters in DESeq2 v1.32.0 per the Bioconductor vignette [Analyzing RNA-seq data with DESeq2 (bioconductor.org)]. Genes with an average TPM (transcripts per million) of less than 1000 across all samples were excluded and those with adjusted p-values $\leq 0.05$ and $|\log_2$ fold-change$| \geq 1$ were considered differentially-expressed. For gene set enrichment analysis, EnrichR tool [94–96] was used and MSigDB Hallmark 2020 gene sets were interrogated. Volcano plots were generated using "ggplot2" package in R. The datasets generated for this study can be found in the NIH repository under accession number PRJNA941341 (https://www.ncbi.nlm.nih.gov/sra/PRJNA941341).

## *In vivo* treatments

Recombinant mIL-33 (Shenandoah, Cat# 200–36) was administered via i.p. (2 μg in a total volume of 200 μL) or i.m. (0.3 μg/muscle in a total volume of 30–50 μL) at the specified time points. Dose and frequency of injections were adapted from reported protocols [10,24,28]. For i.m. treatment, quadriceps, gastrocnemius and tibialis anterior from the same hindlimb received each the dose detailed above. When indicated, i.p. injections also contained 1 μg of recombinant human IL-2 (Gibco, Cat# PHC0021). PBS was used as vehicle.

For treatments with neutralizing antibodies, mice received i.p. injections with 200 μg of anti-IFN-γ clone XMG1.2 (Bio X Cell Cat# BE0055, RRID:AB1107694) or anti-TNF clone XT3.11 (Bio X Cell Cat# BE0058, RRID:AB_1107764) at the specified time points. Control mice received 200 μg of anti-horseradish peroxidase rat IgG1 clone HRPN (Bio X Cell Cat# BE0088, RRID:AB_1107775).

## Body weight and survival evaluation

Total body weight was determined using a precision laboratory balance. Survival of treated mice was monitored every day until experiment ended (21 or 60 dpi).

## Histological analysis

Perfused mouse quadriceps were fixed in formaldehyde solution and embedded in paraffin. Five μm thick sections were stained with activated hematoxylin followed by eosin alcoholic

solution. Histopathological evaluation was performed by a pathologist under light microscopy. Photographs were taken using a Phenoimager Fusion instrument (Akoya Biosciences, RRID: SCR_023274).

## Statistics and graph creation

Unless otherwise indicated, both statistics calculation and graphs creation were performed with GraphPad Prism software (RRID:SCR_002798) version 8.0.1. The normality of data distribution was assessed using Shapiro-Wilk normality test. Statistical significance of mean value comparisons was determined using t-test or One-way ANOVA for normally distributed data, and Mann-Whitney test or Kruskal-Wallis test for non-normally distributed data, as appropriate. Outliers were identified using the ROUT method. P values $\leq 0.05$, considered statistically significant, and P values between 0.05 and 0.1, considered indicative of a tendency, are both indicated in the graphs. P values $> 0.1$ were considered non-significant and are not depicted in the graphs. Data are presented as mean or as mean $\pm$ SEM and the number of animals of each experimental group is indicated in the figure legends or shown in the plots as individual or pooled replicates. Principal Component Analysis graph and volcano plots were generated using "ggplot2" package in R software. Flow cytometry plots were exported from FlowJo software (RRID:SCR_008520) version X.0.7 after data analysis.

## AI Language model assistance

We used ChatGPT (developed by OpenAI) to assist in refining the written content of this study. ChatGPT provided suggestions and corrections based on the input provided by the user, enhancing the clarity and grammar of the text. ChatGPT output was critically revised by the user to ensure it conveys the desired message.

## Supporting information

**S1 Fig. Peripheral target tissues display inflammatory responses during acute *T. cruzi* infection.** Indicators of disease progression and inflammatory response were evaluated in *T. cruzi* infected Foxp3-GFP mice at different days post infection (dpi). (A) Kinetics of total body weight (B) Whole quadriceps muscle (SM) RNAseq data analysis from non-infected (NI) and infected (INF) mice as described in Fig 1G and 1H; N = 3 per group. Volcano plots displaying differentially expressed genes (dots) between INF SM and NI SM. According to Fig 1G, genes associated with interferon gamma response, interferon alpha response and complement pathways are highlighted in red. (C) Kinetics of plasma CPK, CPK-MB, LDH, GOT and GPT activities, and glucose concentration. (D) Kinetics of leukocyte counts in heart normalized to tissue weight (left) and total leukocyte counts in liver (right). In (A and C) data are presented as individual values (circles) while the lines represent the mean. In (D), squares in heart represents values obtained from pools of 4–5 mice, while circles in liver represent values from individual mice, with the lines representing the mean. (A, C and D) Data were collected from 1–3 independent experiments. Statistical significance in (A) was determined by RM one-way ANOVA and P values are relative to 15 dpi, while in (B and C) was determined by one-way ANOVA and P values are relative to 0 dpi. *p $< 0.05$; **p$<0.01$; ****p $< 0.0001$. (TIF)

**S2 Fig. Tregs frequency is reduced in lymphoid and non-lymphoid target tissues during acute *T. cruzi* infection.** Tregs response was evaluated by flow cytometry in spleen, liver, skeletal muscle (SM) and heart from non-infected (NI) and infected (INF) (21 days post infection) Foxp3-GFP mice. (A) Representative dot plots showing the frequency of Tregs (CD4

+ Foxp3-GFP+) within total CD4+ cells from each tissue. (B) Bars displaying Tregs frequency within total CD4+ cells as the mean ± SEM. Circles represent individual mice and squares represent pools of 3–5 mice. Statistical significance was determined by unpaired t test for spleen, liver and SM; and by Mann-Whitney test for heart. P values are indicated in the graphs. (A and B) Data were collected from 3 independent experiments.
(TIF)

**S3 Fig. ST2+ KLRG-1+ Tregs from *T. cruzi* infected mice exhibit a phenotype compatible with *bona fide* trTregs.** Flow cytometry phenotypic analysis of Tregs subsets present in the spleen of non-infected (NI) or infected (INF) (21 days post infection) Foxp3-GFP mice. Representative dot plots showing the expression of each cell marker in ST2+ KLRG-1+ (pink) and ST2- KLRG-1- (gold) Tregs as defined in Fig 2A. Numbers on top right corner of each plot indicate either mean fluorescence intensity or frequency of positive cells for each marker. Data were collected from 3 independent experiments.
(TIF)

**S4 Fig. IL-33 supplementation fails to prevent trTreg reduction in established *T. cruzi* infection.** (A) IL-33 concentration was evaluated in spleen and liver lysates obtained from Foxp3-GFP mice at different days post infection (dpi). Values were normalized to total protein content. Data are presented as individual replicates (circles) and mean (line). (B) Floating bars represent the cumulative counts of live ST2+ KLRG-1+ Tregs (trTregs) and total Tregs after culturing splenic Tregs isolated from non-infected (NI) and infected (INF) Foxp3-GFP mice, as described in Fig 3B. Horizontal lines indicate the average value; N = 2–3 per group. (C) Foxp3-GFPneg CD4+ conventional T cells (Tconv) isolated from the spleen of NI and INF Foxp3-GFP mice were evaluated by flow cytometry. Representative dot plots showing ST2 + KLRG-1+ cells frequency within total Tconv. Left plots correspond to uncultured Tconv, while middle and right plots correspond to Tconv activated with anti-CD3+anti-CD23+IL-2 with or without IL-33 for 72 hours. N = 3 NI and 3 INF. (D and E) Representative plots depicting the frequencies of GFP+ CD4+ Tregs (D and E) and ST2+ KLRG-1+ cells (D) in NI (D) and INF (E) mice receiving intramuscular IL-33. Statistical significance was determined by Kruskal-Wallis test (A), paired t test (B) and Wilcoxon test (E). P values in (A) are relative to 0 dpi: ****p < 0.0001. Data are representative of two (A-C) and one (D and E) independent experiments.
(TIF)

**S5 Fig. Inflammatory and microbial-derived signals are unable to restrict IL-33 mediated expansion of trTregs *in vitro*.** (A) ST2 concentration in plasma, as well as in spleen, skeletal muscle and liver lysates obtained from Foxp3-GFP mice at different days post infection (dpi). Values in tissue lysates were normalized to total protein content and data are presented as individual replicates (circles) and mean (line). Statistical significance was determined by one-way ANOVA. P values are relative to 0 dpi: **p < 0.01; ***p < 0.001. Representative of one experiment. ND: non-detectable. N = 4–13 per group for plasma. (B and C) Representative dot plots showing the frequency of ST2+ KLRG-1+ Tregs (trTregs) within total Tregs isolated from the spleen of non-infected Foxp3-GFP mice and incubated for 72 h with anti-CD3+anti-CD28 together with the addition of different cytokines as follow: IL-2; IL-2+IL-33 and IL2+IL33 plus: microbial ligands (B) or transwell co-cultures or conditioned media providing soluble spleen-derived signals (C), as indicated above each plot. Data were collected from 2 independent experiments. Spl: splenocytes, Tps: trypomastigotes.
(TIF)

**S6 Fig. Early IL-33 administration effect on body weight, SM damage and total Tregs pool.** Infected Foxp3-GFP mice received intraperitoneal IL-33 as schematized in Fig 5A. (A) Total body weight loss between 15 and 21 dpi. (B) Representative Hematoxylin-Eosin stain of quadriceps muscle from both groups (as described in Fig 5D) showing histological details. Blue arrows: necrotic muscle fibers, Orange arrows: muscle fiber with dystrophic calcification, circle: preserved skeletal muscle fiber. Magnification: 40X. (C) Bars displaying Tregs frequency within total CD4+ cells (upper row) and absolute number (bottom row) in skeletal muscle (SM), liver and spleen. For SM, squares represent values obtained from pools of 2–4 mice and cell counts are normalized to tissue weight. For liver and spleen, circles represent values from individual mice. (D) Gating strategy used to identify type 2 innate lymphoid cells (ILC2) as CD45+ CD4- CD8- Lin (CD3, CD19, NK1.1, CD11c)- CD11b- CD127+ ST2+ cells. Dot plots are representative of SM from IL-33-treated infected (INF) mice at 21 dpi. (A and C) Statistical significance was determined by unpaired t test. Data are representative of two (A and B) and three (C) independent experiments.
(TIF)

**S7 Fig. Early IL-33 administration increases trTregs and ILC2 numbers but does not enhance parasite-specific CD8+ T cells at 10 days post infection.** Infected Foxp3-GFP mice received intraperitoneal IL-33 as schematized in Fig 5A and were evaluated at 10 days post infection. (A) Frequency within Tregs and (B) absolute numbers of trTregs in skeletal muscle (SM) and spleen. (C) Frequency within live CD45+ cells and (D) absolute numbers of ILC2 in SM and spleen. (E) Frequency within total CD8+ T cells and (F) absolute numbers of parasite-specific CD8+ T cells in SM and spleen. (G) Violin plots displaying the distribution of plasma GOT, CPK, LDH and GPT activities and glucose concentration. N = 4 per group. (H) *T. cruzi* satellite DNA quantification in SM and spleen. For each tissue, the graphs represent the fold change in parasitic load in IL-33 treated animals relative to the PBS-treated group. (I) Plasma IFN-γ and TNF concentration. (A-F, H and I) Bars represent the mean ± SEM and circles depict individual replicates. For SM, squares represent values obtained from pools of 2–4 mice (A and B) and cell counts are normalized to tissue weight (B, D and F). Statistical significance was determined as follow: Unpaired t test (A-F, H and I), Mann-Whitney test (G). Data are representative of two (A, B, E, F, H, I) and one (C, D, G) independent experiments.
(TIF)

**S1 Table. List of genes associated with inflammatory response and skeletal muscle physiology.** Differentially expressed genes between skeletal muscle from infected (15 days post infection) and non-infected mice are shown for each cellular pathway.
(TIF)

## Acknowledgments

We thank M. P. Abadie, M. P. Crespo, V. Blanco, D. Lutti, C. Noriega, F. A. Frontera, S. R. Oms, R. E. Villarreal, G. Furlán, N. M. Maldonado, A. Romero, L. V. Gatica, M. S. Miró, D. A. Paira and L. Reyna (Centro de Investigaciones en Bioquímica Clínica e Inmunología) for their excellent technical assistance. We acknowledge the NIH Tetramer Core Facility for provision of APC- and Brilliant Violet 421-labeled TSKB20 tetramers. We thank S. Sandrone (Hospital Rawson, Córdoba, Argentina) for her contribution in histopathological analysis. We are also grateful to S. B. Lakshminarayana, C. Osborn, D. Kristen, D. Patra for technical assistance and J. Spector (BioMedical Research, Novartis, United States) for coordinating the global health fellowship.

## Author Contributions

**Conceptualization:** Eva V. Acosta Rodríguez.

**Formal analysis:** Santiago Boccardo.

**Funding acquisition:** Eva V. Acosta Rodríguez.

**Investigation:** Santiago Boccardo, Constanza Rodriguez, Camila M. S. Gimenez, Cintia L. Araujo Furlan, Carolina P. Abrate, Laura Almada, Manuel A. Saldivia Concepción, Peter Skewes-Cox, Srinivasa P. S. Rao, Jorge H. Mukdsi.

**Methodology:** Santiago Boccardo, Eva V. Acosta Rodríguez.

**Project administration:** Carolina L. Montes, Adriana Gruppi, Eva V. Acosta Rodríguez.

**Resources:** Srinivasa P. S. Rao, Carolina L. Montes, Adriana Gruppi, Eva V. Acosta Rodríguez.

**Supervision:** Eva V. Acosta Rodríguez.

**Validation:** Santiago Boccardo, Eva V. Acosta Rodríguez.

**Writing – original draft:** Santiago Boccardo, Eva V. Acosta Rodríguez.

**Writing – review & editing:** Constanza Rodriguez, Camila M. S. Gimenez, Cintia L. Araujo Furlan, Carolina P. Abrate, Laura Almada, Manuel A. Saldivia Concepción, Peter Skewes-Cox, Srinivasa P. S. Rao, Carolina L. Montes, Adriana Gruppi, Eva V. Acosta Rodríguez.

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
