## [Decision Letter · Decision Letter 0]

15 Oct 2024

Dear Dr. Acosta Rodriguez,

Thank you very much for submitting your manuscript "Dynamics of tissue repair regulatory T cells and damage in acute Trypanosoma cruzi infection." for consideration at PLOS Pathogens. As with all papers reviewed by the journal, your manuscript was reviewed by members of the editorial board and by several independent reviewers. In light of the reviews (below this email), we would like to invite the resubmission of a significantly-revised version that takes into account the reviewers' comments.  Reviewer #1 raised substantial questions about the mechanistic correlation between reduction of trTreg cells at the sites of infection and systemic IL-33 diminution. Reviewer #2 raised major concerns particularly regarding the rigor of the data, which need to be considerably improved. Moreover, the three reviewers requested clarification of several major and minor points in the results and discussion sections.

We cannot make any decision about publication until we have seen the revised manuscript and your response to the reviewers' comments. Your revised manuscript is also likely to be sent to reviewers for further evaluation.

Sincerely,

Igor C. Almeida

Guest Editor

PLOS Pathogens

Margaret Phillips

Section Editor

PLOS Pathogens

Michael Malim

Editor-in-Chief

PLOS Pathogens

orcid.org/0000-0002-7699-2064

Reviewer's Responses to Questions

**Part I - Summary**

Reviewer #1: This paper proposes that tissue repairs Tregs might play a role in the immunopathogenesis of infection by T. cruzi. The approach used in initially descriptive and the authors clearly demonstrate a deficiency of trTregs in site of infection accompanied by a systemic deficiency in IL-33. The authors then attempt to correct the deficiency in IL-33 by administering it starting on day 15 pp.i. but observe no change in the numbers of Treg. They then administer IL-15 at disease initiation and do observe an increase in trTreg and an improvement in survival. However, it is difficult to prove that the beneficial effects of IL-33 are mediated via trTreg as marked increases in ILC2s and antigen-specific CD8+ T cells are also observed.

Reviewer #2: This is a well-written manuscript from Boccardo and colleagues that investigates regulatory T cells (Tregs), and in particular, tissue repair Tregs (trTreg), during the course of T cruzi infection and disease. Using a mouse model of T cruzi infection, they demonstrate that Treg and trTreg abundance decreases in several host tissues at peak parasitemia, including the spleen, liver, and skeletal muscle. Because trTreg are defined based on their expression of ST-2, the IL-33 receptor, they test whether provision of IL-33 can restore trTreg abundance in tissues and improve disease outcomes. They found that early IL-33 treatment increased trTreg abundance and parasite-specific CD8 T cell responses, which correlated with improved disease outcome and lowered parasite burden. Thus, the findings point to a potential therapeutic potential of IL-33 treatment in context of T cruzi infection.

In general, the data support the claims, but there are several places where the rigor of the data could be improved (outlined below). In addition, there are a few key places where further discussion is needed, as noted below. However, overall the study is well-done and the findings are interesting – well done!

Major critiques:

1. I find it odd that IL-33 supplementation is associated with an increased abundance of trTreg given that there are so few ST-2+ Treg present. The authors should speculate as to how IL-33 is able to increase the % or # of ST-2+ trTregs without the cells expressing the receptor for the cytokine. Presumably this is through an indirect mechanism, or do they propose that this is due to proliferation of a few ST-2+ Tregs or perhaps conversion of conventional CD4 T cells expressing ST-2 into Foxp3+ trTreg?

2. Rigor of the data needs to be improved:

a. In general, show all the data points rather than just mean and SEM (for example, in Fig 1A-C and 2B). This will help to clarify how many mice are included in each experiment, as it is not always stated in the figure legends.

b. Please show representative flow cytometry dot plots (and not just histograms as shown in Fig S3) for Fig 2C. There are clearly very few trTreg in the spleen (as shown in Fig 2A-B), calling into question the reliability of flow cytometry data looking at activation markers on this very small subset of cells. How many trTreg were used per tissue for this downstream analysis shown in Fig 2C? Showing the numbers of trTreg and KLRG1-ST2- Treg would be informative in this case, so that we can be sure that the % of trTreg expressing all the of markers shown in Fig 2C (as compared to KLRG1-ST2- Treg) are not near zero due simply to the lack of trTreg present in the tissue at 21 days post-infection.

c. There are several places where flow plots are shown without quantification, so it appears that there is only an N of 1 (for example, Fig 3B and Fig 4A). Technical or experimental replicates should be shown with statistical analysis for increased rigor. In many cases, differences that appear very minor are emphasized, and a statistical analysis could perhaps help to back up the author’s claims (for example, differences in Fig 4A are very slight).

3. Survival experiments need to be compared between figures and the differences in control groups need to be accounted for. In figure 4C, the control group has 100% survival, whereas in figure 5B, it looks as though apx 50% of the control mice succumb to infection. This is important because it calls into question the significance of the findings. Are there differences in the control groups between the two experiments that could account for these major differences? One way to reconcile this would be to show survival curves and % change in body weight (relative to day 0) following infection in Figure 1 to establish the model.

4. The authors show in Fig 5 that provision of IL-33 early after infection results in an increased frequency of Ag-specific CD8 T cells in the SM, liver, and spleen (Fig 5I), yet no difference in plasma IFNg and a decrease in plasma TNFa (Fig 5L). To see if there is a functional difference in the CD8 T cells, it could be useful to perform intracellular cytokine staining to look at cytokine expression on a single-cell level, or perhaps measure tissue lysates for cytokines. Alternatively, can the authors speculate on how IL-33 supplementation might be reducing the parasite burden (Fig 5K)?

5. It would be nice for the authors to include some histology data to show a decrease in tissue pathology associated with an increase in trTreg abundance following IL-33 treatment. This would firmly establish that trTreg are indeed “tissue repair” Tregs beyond use of phenotypic markers.

6. It would be interesting to try i.m. IL-33 supplementation early after infection or perhaps even pre-infection to see if that has a stronger effect on trTreg in skeletal muscle. However, I do not consider that experiment to be essential for publication of this manuscript. Or was IL-33 delivered i.m. in the experiment shown in Figure 5 – the delivery route was not clear from the figure legend.

Minor critiques:

1. In the introduction, the authors refer to trTreg in many places where the citing literature was studying tissue Treg. Please take care to clearly distinguish analyses of tissue Tregs in general from trTreg (as not all tissue Tregs have a trTreg phenotype). As an example, in line 66-69, the statement seems to refer more broadly to tissue Treg, although the authors attribute the statement to trTreg. There is a similar problem with lines 72-74.

2. There is a typo in lines 207-209 – the data in Fig 2C do not show that spleen trTreg from non-infected mice exhibit higher expression levels of most markers compared to ST2-KLRG1- Treg but rather the opposite.

3. The sentence in lines 220-222 is unclear; in particular, the phrase “presenting particularities likely as a consequence of the infection”. Please revise to clarify what you mean. Furthermore, please clarify what you mean by “bone fide trTreg”.

4. Line 250: please fix typo “Ssplenic”.

5. In Figures 1C and 1E, please clarify in the figure legend how the number of CD45+ cells in tissue was quantified (was this flow cytometry, or microscopy, or something else?).

6. Please indicate the limit of detection for IL-33 detection in Fig 3A.

7. In Fig S4B there appears to be a typo – I believe this is looking at CD4+Foxp3GFP-negative cells, yet the figure indicates “Tregs gate”.

Reviewer #3: The data presented are indeed relevant within the context of acute T. cruzi infection in the animal model. However, the results appear somewhat controversial compared to previous findings from the same group regarding the role of conventional Treg cells. In their earlier work, it was observed that depletion of this Treg population led to an increase in parasite-specific CD8+ T cells (as measured using e TSKB20 and TS peptide) and a decrease in parasitemia.

In the current study, while IL-33 is shown to promote the expansion and activation of regulatory T cells, you report an increase in the same population of parasite-specific CD8+ T cells alongside a decrease in parasite load in tissues, which presents opposing effects. Although you have made attempts to explain these controversial results, further elaboration may be necessary to elucidate a possible physiological mechanism.

The data presented in Figure 2 are expressed as mean ± SEM. However, I would like to clarify the methodology regarding the normalization of cell counts for the SM tissue. The text states that cell counts are normalized to tissue weight and that all values were obtained from pools of three mice, with N = 2-7 per day post-infection (dpi). If there were only two mice per dpi, could you please elaborate on how a pool of three mice was achieved?

Regarding Figure 3D, it appears to depict flow cytometry results for trTreg within Treg populations. However, the description as "Flow cytometry analysis of Tregs present in..." may lead to some confusion, particularly since no frequency or parameters for this population are calculated. I suggest rephrasing this section for clarity.

Additionally, it may be beneficial to include data corresponding to non-infected (NI) animals in a manner similar to that of infected mice, rather than presenting it as a line of media value. This could provide more context for the findings.

It is somewhat unclear why both individual samples and pooled samples were used for the same tissue type, such as SM. Could you provide insight into this methodological choice?

I would appreciate some clarification on why it was not feasible to dilute the liver sample to fall within the assay's dynamic range. Understanding this limitation would be helpful.

Parasite Load Measurement

Did you measure the parasite load in mice treated with IL-33 alone, as well as in those treated with both IL-33 and IL-2?

Furthermore, do you have data on mice treated with IL-33 and IFN-gamma? This information would be valuable to determine if there is a direct correlation between the TH1-like response and the effect of IL-33 on rtTreg.

did you determine the frequency of Treg cells across the different treatment groups? This information would enhance our understanding of the immune response in your study, and is in relation with my previous argumentation.

**Part II – Major Issues: Key Experiments Required for Acceptance**

Reviewer #1: The authors clearly show a diminution in trTreg at the sites of infection and a diminution in systemic IL-33. This finding appears solid but the never directly demonstrate that the two observations are mechanistically related. There are several areas of the manuscript that need to be addressed:

1. The authors contend that the trTreg from infected mice are intrinsically capable of responding to IL-33 based on their capacity to expand in vitro in response to anti-CD3/CD28/IL-2/IL-33 administration (fig. 3B). This study is performed starting with a total Treg population. The authors need to show absolute counts of Total Treg and trTreg at the end of the culture to prove actual expansion. It is also very difficult to conclude that this in vitro study using maximal stimulatory conditions has any relevance to the lack of response seen in vivo.

2. The authors then postulate that the presence of IFNg is responsible for the failure of the trTregs to expand in vivo. The experiment shown in fig. 4A is far from convincing. It is again based on the in vitro assay and the purported suppression of expansion (7.9-6.1%) is surely not suppressive. This is a single experiment without statistical variation. The authors show that TNF appears to play a beneficial effect in survival, but the studies with anti-IFNg are not carried out long enough to observe changes from controls (cf.fig. 4c and 5b). The claimed increases in the percentages and absolute numbers of trTreg after IFNg treatment are very small and likely to mediate any beneficial effects.

3. The studies on the early administration of IL-33 which do show a beneficial effect using multiple parameters are impressive, but certainly do not prove that the effects are secondary to the increase in trTreg. As the authors point out an early increase in ILC2s also occurs and there are likely many more ILC2s than trTreg in vivo. The only way to dissect the contribution of ILC2s is to delete them for example by using GATAfl/flXKLG1 mice or other strains deficient in ILC2s. The contribution of CD8+ T cells could easily be tested by using depleting antibodies a widely accepted technique.

Reviewer #2: See above

Reviewer #3: (No Response)

**Part III – Minor Issues: Editorial and Data Presentation Modifications**

Reviewer #1: Nonew

Reviewer #2: See above

Reviewer #3: Minor details,

Line 250: Ssplenic, erase S

Line 1231, missed space

Figure S1: missed u in Glucose

PLOS authors have the option to publish the peer review history of their article (what does this mean?). If published, this will include your full peer review and any attached files.

Reviewer #1: No

Reviewer #2: No

Reviewer #3: No
---

## [Decision Letter · Decision Letter 1]

14 Jan 2025

Dear Dr. Acosta Rodriguez,

We are pleased to inform you that your manuscript 'Dynamics of tissue repair regulatory T cells and damage in acute Trypanosoma cruzi infection.' has been provisionally accepted for publication in PLOS Pathogens.

Best regards,

Igor C. Almeida

Guest Editor

PLOS Pathogens

Margaret Phillips

Section Editor

PLOS Pathogens

Sumita Bhaduri-McIntosh

Editor-in-Chief

PLOS Pathogens

orcid.org/0000-0003-2946-9497

Michael Malim

Editor-in-Chief

PLOS Pathogens

orcid.org/0000-0002-7699-2064

Reviewer 3 recommended that the authors include, as a Supplementary Figure, data demonstrating that the parasite-specific CD8 T cell response did not increase following delayed IL-33 or mIL-2 + IL-33 administration. To avoid delaying the editorial decision on the acceptance of this manuscript, the authors are advised to include the requested supplementary figure in the revised version of the accepted manuscript.

Reviewer Comments (if any, and for reference):

Reviewer's Responses to Questions

**Part I - Summary**

Reviewer #1: (No Response)

Reviewer #3: No further remarks

**Part II – Major Issues: Key Experiments Required for Acceptance**

Reviewer #1: (No Response)

Reviewer #3: No further remarks

**Part III – Minor Issues: Editorial and Data Presentation Modifications**

Reviewer #1: (No Response)

Reviewer #3: I suggest the authors to include as Supplementary Figure, the data demonstrating that parasite-specific CD8 T cell response did not increase after delayed Il-33 or mIl-2+IL-33 administration.

PLOS authors have the option to publish the peer review history of their article (what does this mean?). If published, this will include your full peer review and any attached files.

Reviewer #1: No

Reviewer #3: No

---

## [Editor Report · Acceptance letter]

24 Jan 2025

Dear Dr. Acosta Rodriguez,

We are delighted to inform you that your manuscript, "Dynamics of tissue repair regulatory T cells and damage in acute Trypanosoma cruzi infection.," has been formally accepted for publication in PLOS Pathogens.

Best regards,

Sumita Bhaduri-McIntosh

Editor-in-Chief

PLOS Pathogens

orcid.org/0000-0003-2946-9497

Michael Malim

Editor-in-Chief

PLOS Pathogens

orcid.org/0000-0002-7699-2064